# Metal microdrive and head cap system for silicon probe recovery in freely moving rodent

**Mihály Vöröslakos[1†], Peter C Petersen[1†], Balázs Vöröslakos[2†], György Buzsáki[1,3]\***

[1]Neuroscience Institute, New York University, New York, United States; [2]Budapest University of Technology and Economics, Faculty of Mechanical Engineering, Budapest, Hungary; [3]Department of Neurology, Langone Medical Center, New York University, New York, United States

**Abstract** High-yield electrophysiological extracellular recording in freely moving rodents provides a unique window into the temporal dynamics of neural circuits. Recording from unrestrained animals is critical to investigate brain activity during natural behaviors. The use and implantation of high-channel-count silicon probes represent the largest cost and experimental complexity associated with such recordings making a recoverable and reusable system desirable. To address this, we have designed and tested a novel 3D printed head-gear system for freely moving mice and rats. The system consists of a recoverable microdrive printed in stainless steel and a plastic head cap system, allowing researchers to reuse the silicon probes with ease, decreasing the effective cost, and the experimental effort and complexity. The cap designs are modular and provide structural protection and electrical shielding to the implanted hardware and electronics. We provide detailed procedural instructions allowing researchers to adapt and flexibly modify the head-gear system.

**\*For correspondence:**
gyorgy.buzsaki@nyulangone.org

[†]These authors contributed equally to this work

**Competing interests:** The authors declare that no competing interests exist.

## Introduction

Action potentials are the common currency of communication between neurons and they can be detected as voltage fluctuation in the extracellular space (*Adrian and Moruzzi, 1939*). However, recording from representative ensembles of neurons simultaneously requires electrodes with multiple recording sites. Multi-wire twisted electrodes (tetrodes) and silicon probes offer the possibility to record tens to hundreds of neurons simultaneously from multiple cortical and subcortical structures simultaneously in freely moving animals (*McNaughton et al., 1983*; *Wise and Najafi, 1991*; *Csicsvari et al., 2003*; *Buzsáki, 2004*; *Blanche et al., 2005*; *Montgomery et al., 2008*; *Jun et al., 2017*). For cost benefits, microwire arrays are a popular choice for neuroscientists, despite the amount of manual labor involved (*Edell et al., 1992*). Silicon probes, while more expensive, do not require assembly, the tissue-volume displacement is minimal (*Buzsáki, 2004*; *Kipke et al., 2008*), recording properties are consistent (site impedance and locations) and geometric configurations (number of shanks, distance, and pattern of recording sites) can be customized to suit the architecture of the particular brain structure under study (*Wise and Najafi, 1991*; *Scholvin, 2016*). The affordability of high-channel-count electrophysiology amplifier chips (e.g., RHD-2132 and RHD-2164, Intan Technologies, Los Angeles, CA; *Berényi et al., 2014*) and integrated designs (*Jun et al., 2017*) have accelerated the spread of large-scale recordings. Integration of µLEDs into silicon-based electrodes and can offer unique spatiotemporal control of neuronal activity (*Wu et al., 2015*; *Kim et al., 2020*).

The most expensive component of the head gear is the recording probe (from $1000 for 32-channel passive recording probes to > $3000 for µLED probes or larger channel count probes).

Therefore, reusing silicon probes in freely moving animals is an important current goal (*Juavinett et al., 2019*). In addition to reducing costs, repeated usage of the same probe/headgear would allow for better consistency in recordings across animals, enhances data reproducibility, and would reduce electrode/headgear preparation for surgery. Achieving this goal requires an integrated design of a reusable microdrive and head gear to increase recording stability and protect/shield sensitive drive and electronic components (*Chung et al., 2017*; *Senzai et al., 2019*).

Currently available headgear systems are typically large, reducing the ability to record from multiple brain structures in mice. In contrast to recent progress in recording electrodes, the development of implantation techniques such as the Flexdrive, Shuttledrive, DMCdrive and the Hyperdrive (*Voigts et al., 2013*; *Voigts et al., 2020*; *Kim et al., 2020*; *Lu et al., 2018*) has lagged behind. Electrodes are either fixed in brain tissue or attached to a microdrive to allow the advancement of the electrode after implantation (*Chung et al., 2017*; *Fee and Leonardo, 2001*; *Korshunov, 2006*; *Vandecasteele et al., 2012*; *Wilson and McNaughton, 1993*; *Yamamoto and Wilson, 2008*). Microdrives and accompanying head gear protection and shielding inevitably add extra weight (weight = 0.12 g - 1 g, drives designed for mice) and volume (skull surface area = 7.68–252 mm$^2$, drives designed for mice) to the implant (*Table 1*). The weight, volume and footprint of the microdrive can limit comfortable movement of small rodents and can prevent flexible multiregional recordings in mice (*Headley et al., 2015*). Yet, chronic recordings from freely behaving subjects are essential in many experiments, where the relationship between neuronal activity and movement, perception, learning and memory, decision making, and other forms of cognition are studied to disambiguate overt behavior and hidden variables (*Juavinett et al., 2019*; *Jun et al., 2017*; *Steinmetz, 2020*). An ideal microdrive should have high precision movement, mechanical stability, minimal size, low weight, and the ability for flexible customization. Commercially available microdrives are expensive and hard to customize. Disposable 3-D printed customized drives and head gear have reduced costs (*Headley et al., 2015*; *Chung et al., 2017*; *Allen et al., 2020*). Most importantly, recovery and reimplantation of recording probes are limited with currently available headgears.

Below, we report the design and testing of an integrated 3D printed headgear system (including microdrives and protective head cap) that is adaptable for multiple recording devices for both mice and rats. Our design reduces surgery time substantially and the small footprint of the metal Microdrive allows targeting multiple brain structures. The fast and reliable recovery of the probe and reuse of the same system in multiple animals decreases costs and experimenter effort.

## Results

### Recoverable metal microdrive

3D printing has taken science and industries by the storm, offering in-house design customization, fast iterative development, and cheap production using professional printers based on filament extrusion (e.g., MakerBot Industries, New York, NY) and liquid resin (e.g., Form three by Formlabs, Sommerville, MA). Yet, plastic prints have limitations mostly due to the low strength of the materials.

**Table 1.** Summary of microdrive designs used in mice.

| Study/Company | Width (mm) | Length (mm) | Height (mm) | Footprint (mm$^2$) | Weight (g) | Travel distance (mm) | Easy recovery |
|---|---|---|---|---|---|---|---|
| *Vandecasteele et al., 2012* | 4.3 | 6.4 | 13 | 27.52 | 0.6 | 8–12 | no |
| Janelia Research Campus | 3.5 | 3.8 | 9 | 13.3 | 0.8 | 5 | no |
| Janelia Research Campus | 2.5 | 3.8 | 10 | 9.5 | 0.5 | 5 | no |
| Cambridge Neurotech | 2.5 | 4 | 9 | 10 | 0.54 | 5 | no |
| Neuronexus | 12.5 | 11.5 | 8.5 | 143.75 | 0.36 | 1 | no |
| NeuroNex MINT | 3.2 | 7.5 | 16 | 24 | 0.67 | 4.8 | yes |
| *Chung et al., 2017* | 6.26 | 5.26 | 9 | 32.93 | 0.4 | 2 | yes |
| *Juavinett et al., 2019* | 18 | 14 | 20 | 252 | 1 | - | yes |
| Voroslakos et. al. | 3.1 | 5 | 15.3 | 15.5 | 0.87 | 7 | yes |

Repeated use of plastic threads results in rapid deterioration, which can be prevented by metal-to-metal connection (*Figure 1—figure supplement 1*). Recently, metal printing has become affordable offering increased strength, with options for printing in aluminum, stainless steel and even titanium with similar printing resolution to plastics. Here, we have taken advantage of this and constructed a 3D printed microdrive from stainless steel (stainless steel 316L, 20 µm resolution), which offers superior strength and form factor compared to plastic prints (Young's modulus of stainless steel: ~180 GPa vs plastic: ~2 GPa). The metal printing allows reuse of the drives with minimal wear, driving the effective cost down.

The microdrive is composed of three metal parts: an arm, a body, and a base (*Figure 1A and B*) and has a footprint of 15.5 mm$^2$ (width: 3.1 mm, length: 5 mm, height: 15.3 mm). The detachable base allows for easy recovery of probes. The arm/shuttle is mounted on a screw to the drive body, allowing it to move linearly along the vertical axis simply by turning the screw (270 µm/ turn). The constructed microdrive has a total travel distance of 6 mm (maximum distance between arm and bottom of the drive body), allowing one to record from multiple brain regions across days and weeks. Due to its small form factor, multiple probes can be implanted in the same animal (*Figure 1C*). The drives come with a stereotaxic implantation tool, printed in plastic (clear v4 resin from FormLabs), for user-friendly and reliable implantations and probe recovery (*Figure 1—figure supplement 2*), consisting of a stereotactic manipulator attachment and a microdrive holder (*Figure 1D and E*, *Figure 1—video 1* and *2*).

The fully assembled steel microdrive weighs 0.87 g (base: 0.23 g, shuttle/arm with nut: 0.16 g, drive body with screw and metal bar: 0.49 g). This weight and dimensions are similar to other commercially available or custom-made electrode microdrives (*Table 1*). The design files for the microdrive can be submitted to commercial 3D printing companies (e.g., Proto Labs, Maple Plain, MN, https://www.protolabs.com/) allowing for high-quality printing and fast production. The printing costs of the three components can be as low as $170 (base: $50, body: $60, arm: $55), a highly competitive price compared to commercial microdrives.

## Inclusion criteria: microdrive systems that used silicon probes in freely moving mice

### Mouse cap

To reuse silicon probes in multiple experiments, both the microdrive and the head cap have to be sturdy, easy to disassemble and reassemble. The mouse cap is composed of three parts: a base, a left-side wall, and a right-side wall (*Figure 2A*). The cap-base is attached to the skull of the animal during anesthesia using a ring of Metabond cement, serving as a base for the rest of the cap. There is no need for skull support screws, making the head cap minimally invasive. The cap has a large internal window shaped as an elongated octagon, following the outer ridge of the skull, giving wide access for various surgical needs (*Figure 2B*). The sidewalls provide structural support, electrical shielding (by acting as a Faraday cage), and physical protection of the silicon probes, hardware, and electronics. The internal volume allows for great flexibility and can fit two Omnetics preamplifier-connectors, as well as optic fibers. The sidewalls attach to the base using a rail and with three support screws (*Figure 2—video 1*).

The entire cap weighs 2.2 g (base: 0.19 g, walls with male header pins and copper mesh: 0.98 g each, and 000–120 screws: 0.05 g; *Figure 2C*). A chronically implanted mouse can carry this cap with one (or more) implanted silicon probe and with a custom connector for electrical stimulation (*Figure 2D*). High-quality electrophysiological signals can be collected from freely moving mice for weeks (*Figure 2E and F*). The system can be customized as needed, using our CAD files (see Methods section). We recommend printing the cap system on the Formlabs Form 2/3 resin printer or a comparable 3D printer (requires 25–50 µm resolution).

### Rat cap

The typical Long-Evans rat is approximately ten times heavier than the mouse (~400 g), and requires a sturdier cap system, capable of withstanding forceful impacts and provide increased protection of the electronics and hardware. The rat cap is composed of four parts: a base, a left-side wall, a right-side wall, and a top cover (*Figure 3A*). The octagon-shaped base aligns with the outer rim of the rat's dorsal skull surface and is attached with Metabond cement, with no need for skull support

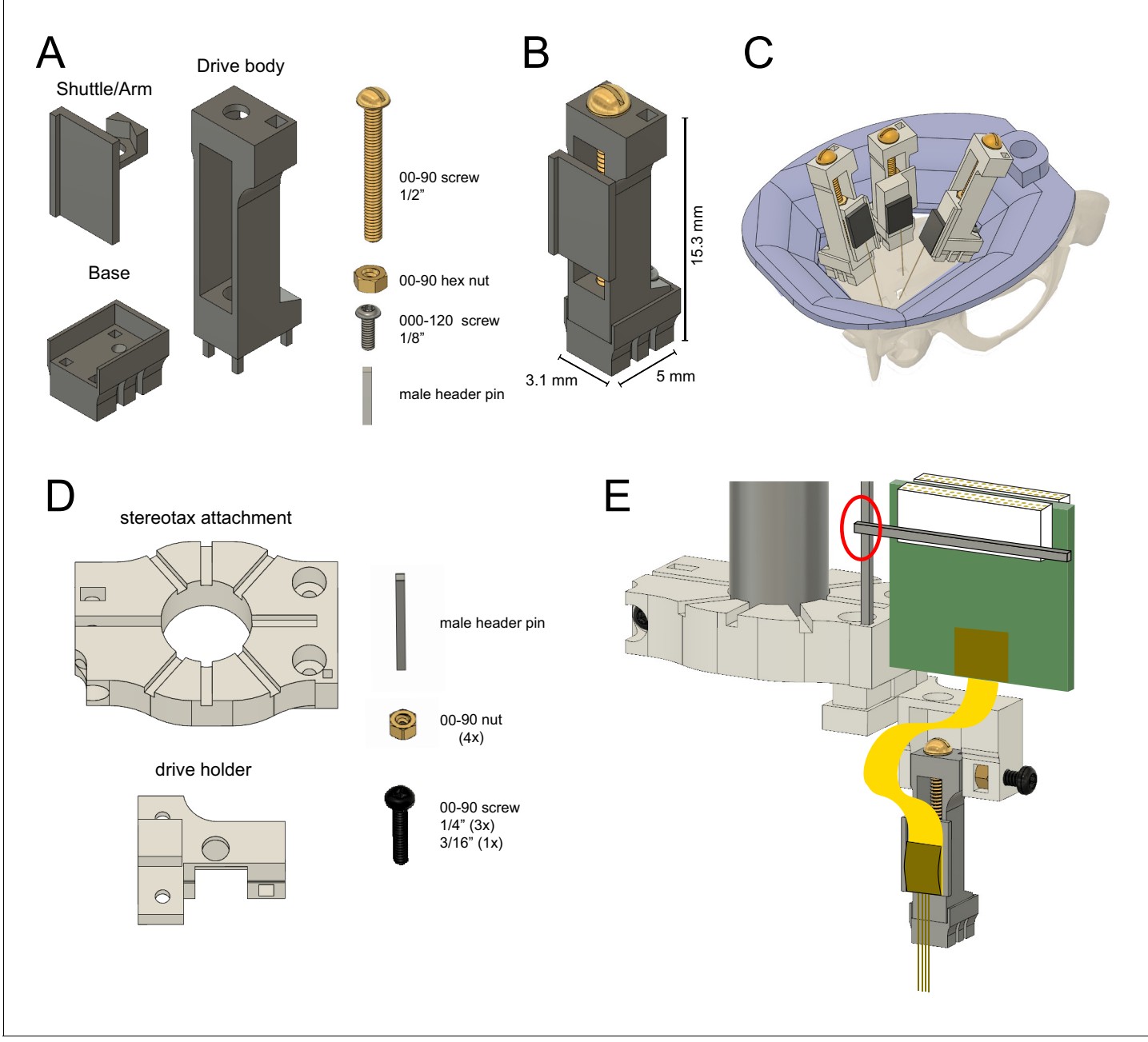

**Figure 1.** Reusable metal microdrive. (**A**) The metal microdrive consists of three main parts: a drive body, a movable arm/shuttle, and a removable base. All components are 3D printed in stainless steel. Additional necessary components are a 00–90, 1/2 'brass screw, a 00–90 brass hex nut, a 000–120, 1/8' stainless steel screw fixing the drive to the base, and a male header pin. (**B**) The assembled drive with dimensions. (**C**) Schematic showing three microdrives, with silicon probes attached, implanted in a rat to target hippocampus, medial and lateral entorhinal cortices. 3D printed resin head cap is shown in purple. (**D**) 3D printed stereotaxic attachment and drive holder together with assembly pieces: male header pin, four 00–90 brass hex nuts, three 00–90, 1/4' and a 3/16' stainless steel screw. (**E**) Stereotaxic attachment with the metal drive assembled, and a probe attached, ready for implantation (red circle highlights the temporary soldering joint for the Omnetics connector). The backend of the silicon probe is attached to the arm using cyanoacrylate glue.

The online version of this article includes the following video and figure supplement(s) for figure 1:

**Figure supplement 1.** Developmental stages of metal, recoverable microdrive.

**Figure supplement 2.** Internal lab survey using recoverable, plastic microdrives.

**Figure 1—video 1.** Assembly of metal microdrive.

https://elifesciences.org/articles/65859#fig1video1

**Figure 1—video 2.** Neuropixels probe attachment to metal microdrive.

*Figure 1 continued on next page*

screws, making it minimally invasive (*Figure 3—figure supplement 1A*). The two side walls are attached to the base with a single rotation-axis located in the front of the base, attached with a long screw (*Figure 3B*, top part). The walls are held in place on the base, using a rail and two screws in the back. The sidewalls have two sets of male header pins for soldering standard Omnetics probe connectors (see Surgical Instructions). The lid can be locked with a thumb screw and has holes for air ventilation (*Figure 3B* bottom part, *Figure 3—video 1*). High-quality electrophysiological signals can be collected from freely moving rats for weeks either using silicon probes (*Figure 3C and D*) or Neuropixels probes (*Figure 3—figure supplement 2*).

For more complex experiments, the cap system can be modified to increase the available skull surface (*Figure 3—figure supplement 1A and B*). This modified base is held by bone screws implanted in the temporal bone and covered with dental cement (*Figure 3—figure supplement 1B*

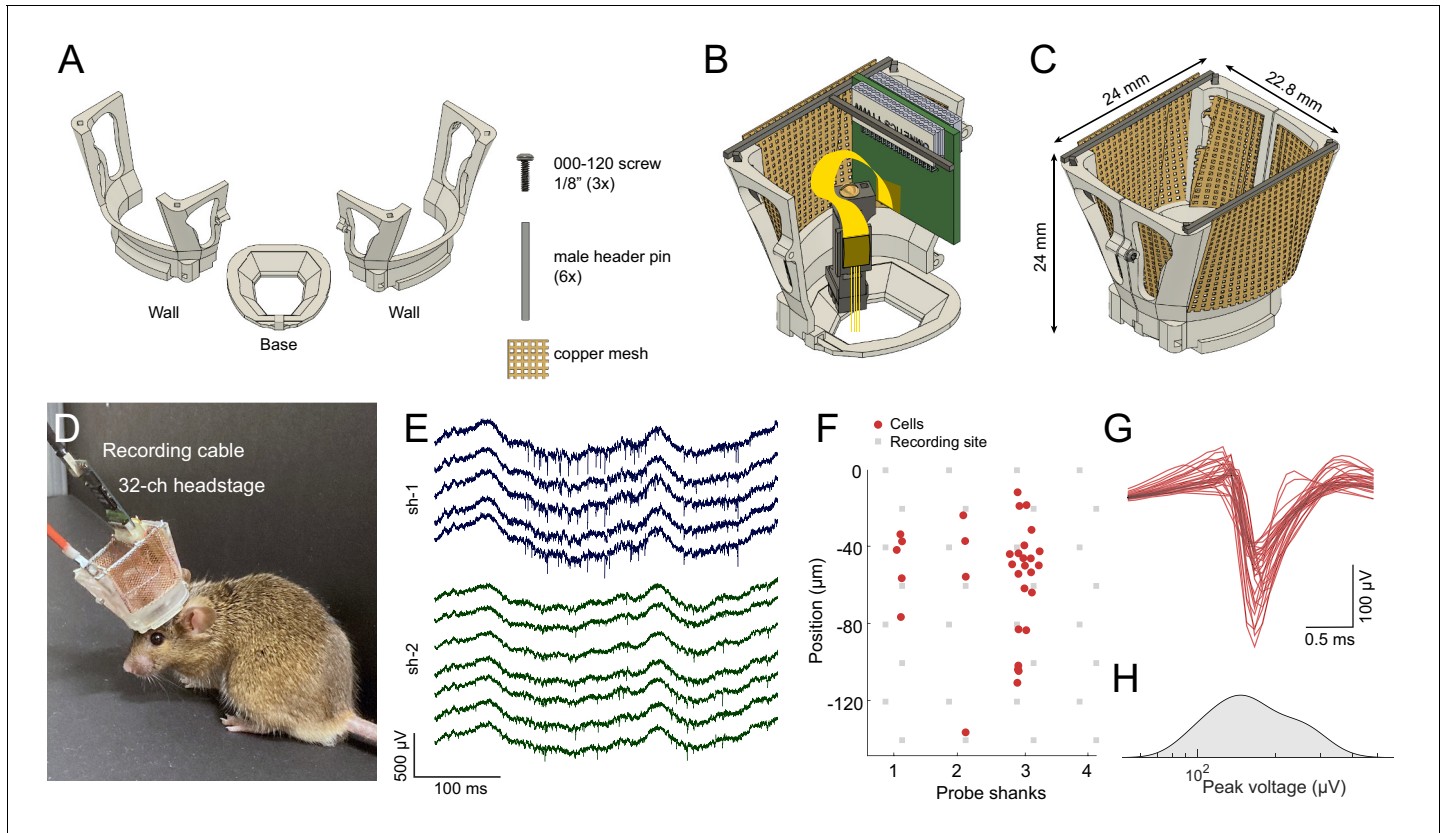

**Figure 2.** Mouse cap. (**A**) The mouse cap consists of three main 3D printed parts: a base, and two side walls. The pieces are assembled with three 000–120, 1/8' steel screws, six male header pins, and copper mesh. (**B**) The base with the left side wall attached. Copper mesh was attached in three pieces to the wall, and a male header pin was soldered across the top of the wall. (**C**) The fully assembled mouse cap. (**D**) The implanted headgear with preamplifier and recording cable attached. (**E**) Wide-band extracellular traces recorded from the prelimbic cortex of the implanted mouse shown in (**D**) using a multi-shank silicon probe during food pellet chasing exploration (sh-1 and sh-2 denote shank 1 and shank 2 of the silicon probe). (**F**) Well-isolated single units can be recorded in the anterior cingulate cortex using the mouse cap system and microdrive (n = 31 putative single units with a 4-shank probe; same session as in E). The location of the maximum waveform amplitude of each neuron is shown (0 µm corresponds to the location of the topmost channel of the shank). (**G**) The average spike waveform for each neuron. (**H**) Distribution of the spike amplitude across the recorded neurons (n = 31).

The online version of this article includes the following video for figure 2:

**Figure 2—video 1.** Assembly of mouse cap.

https://elifesciences.org/articles/65859#fig2video1

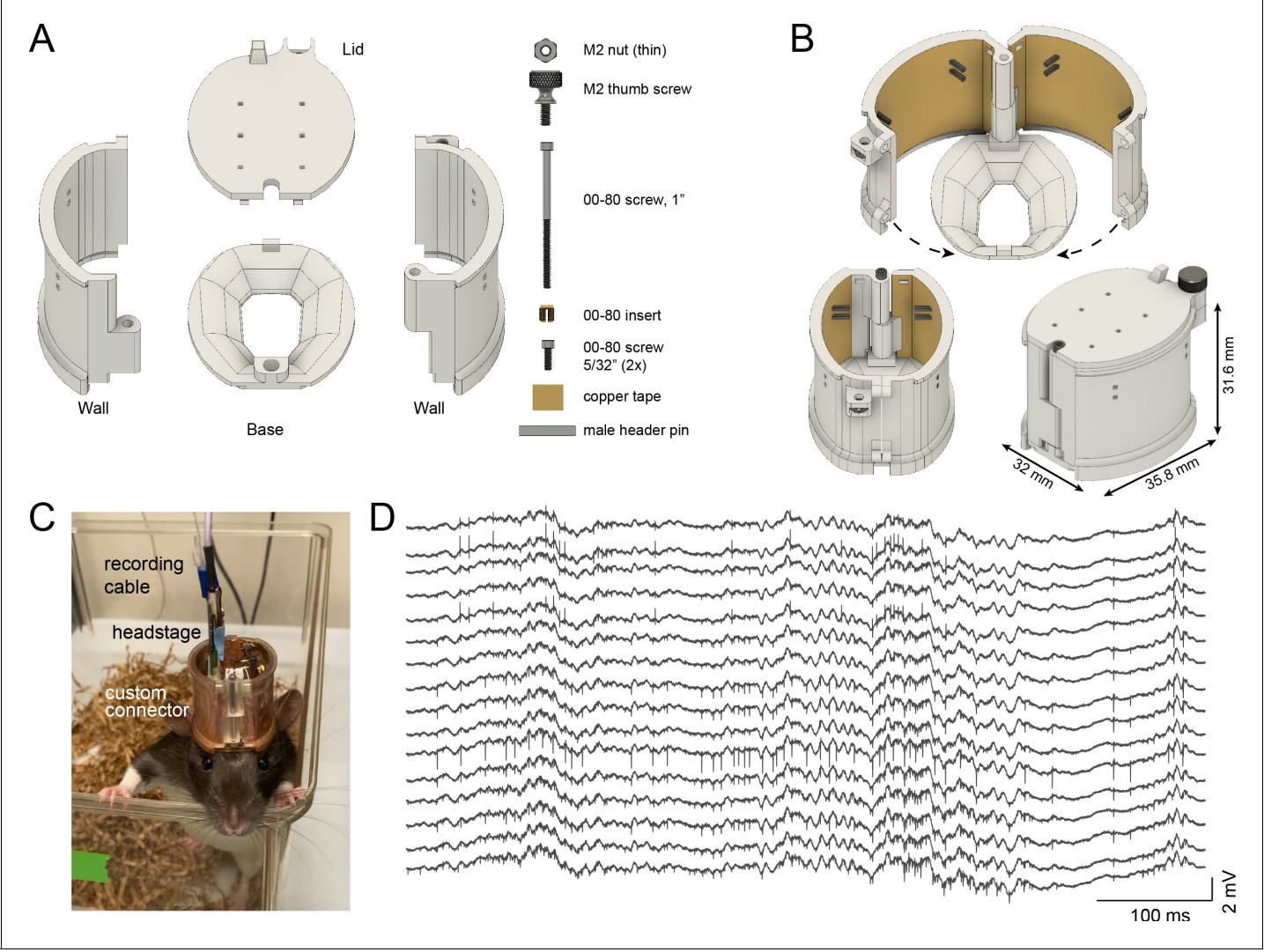

**Figure 3.** Rat cap. (**A**) The rat cap consists of four main 3D printed plastic parts: a base, two side walls, and a lid. To assemble the components, an M2 nut, M2 thumb screw, a 00–80, 1' screw, a 00–80 insert, and two 00–80, 5/32' screws are also needed. (**B**) The assembled rat cap is shown with sidewalls in an open position (top image), closed configuration without (bottom left) and with the lid in place (bottom right). (**C**) A Long-Evans rat in its home cage with the rat cap, connected to preamplifier and cable. (**D**) Extracellular traces recorded on post-op day 18 from the same animal.

The online version of this article includes the following video and figure supplement(s) for figure 3:

**Figure supplement 1.** Headcap and microdrive customization.

**Figure supplement 2.** Implantation of Neuropixels probe in a rat using metal microdrive and rat cap system.

**Figure 3—video 1.** Assembly of rat cap.

https://elifesciences.org/articles/65859#fig3video1

**Figure 3—video 2.** Assembly of the 3D-printed head cap.

https://elifesciences.org/articles/65859#fig3video2

**Figure 3—video 3.** Assembly of the copper mesh head cap.

https://elifesciences.org/articles/65859#fig3video3

right part). Increasing the inner volume of the cap system and using metal recoverable microdrives enable multiprobe implantations (*Figure 3—figure supplement 1C*).

The entire design weighs 11.03 g (base: 1.04 g, right wall with male header pins and copper tape: 3.48 g, left wall with male header pins and copper tape: 3.68 g, top with thumb screw: 2.35 g and 00–80 screws: 0.48 g; *Figure 3B* bottom, right).

## Surgical advantages using the head cap systems

The modular system decreases the duration of the surgery and allows for faster post-operative recovery of the animal, due to four important modifications. 1. The head cap is prepared before surgery and can be reused easily. 2. The cap does not need support screws, reducing the invasiveness of the surgery and accelerating the animal's recovery. 3. The 3D printed cap-base is secured with a single step, by attaching it to the dorsal surface of the skull with Metabond cement. This ensures alignment precision relative to the brain surface, easier probe recovery, and reusability. 4. The electric shielding and structural support is implemented in the reusable head cap, decreasing extra manual steps for the construction of the protective cap from copper mesh, male header pins and grip cement during surgery (*Vandecasteele et al., 2012*).

These steps offer a time savings from 40 to 90 min (*Figure 3—videos 2* and *3*), compared to a manually constructed cap during surgery (*Vandecasteele et al., 2012*).

Further, the modular cap system substantially increases flexibility during an implantation procedure. Because the sides can easily be disassembled and reassembled, complex surgical procedures can be split into multiple sessions when needed. In the first session the skull is prepared, and the base of the cap is attached to the skull. After recovery, the craniotomy and implantation are performed in a second surgery. This double-step procedure results in a speedy recovery of the animal and reduces the likelihood of human error during extended procedures. Additionally, subsequent troubleshooting can be performed through the course of long chronic experiments with minimal disruption to the animal and the implanted components.

## Probe recovery

To recover the probe at the end of a chronic experiment, the drive holder is aligned with the drive using the stereotactic frame. Once the position is aligned in the x-y plane, the drive holder is moved downwards (*Figure 4A*, step 1). Next, the top of the drive is secured with the screw located on the side of the drive holder (*Figure 4A*, step 2). The 000–120 screw is removed from the base (*Figure 4B*, step 1) and the drive is moved upwards carefully (*Figure 4B* step two and C). We recommend to carefully monitor the shanks of the probe under a microscope during the entire recovery procedure and, if any unexpected movement of the probe is observed, return to the previous step to make sure that everything is secured properly (*Figure 4—videos 1* and *2*).

The removed silicon probe (NeuroNexus, Cambridge Neurotech, Diagnostic Biochips products; Neuropixels) is cleaned by initially rinsing it in distilled water, then contact lens solution (containing protease) and distilled water again; each washing step should last for at least 12 hr. Soak the silicon shanks only (avoid soaking the backend area). If extra tissue or debris is detected between the shanks, it can be carefully removed by a fine needle (26 gauge or smaller) under a microscope. After recovering Neuropixels 1.0 probes, the probe shank should be soaked in 1% tergazyme (Alconox) for 24–48 hr, then rinse in distilled water and isopropyl alcohol (*Luo et al., 2020*).

## Quantification of single unit quality measures

Microdrives allow experimenters to record from novel sets of neurons in successive sessions, surveying thousands of neurons from the same structure in a single animal (*Girardeau et al., 2017*). With the recoverable metal microdrive, we recorded across several days from the same animal while adjusting the implantation depth (500 µm to 780 µm) across days (*Figure 5A–F*). Across days of recordings, while the probe was moved to record from different depths, the unit yield increased (*Figure 5H*), the waveform amplitude increased (*Figure 5E*) while the relative noise level decreased (*Figure 5F*), suggesting either that the distance between the electrode sites and neuron bodies decreased or that large size neurons were recorded.

## Effect of head gear type on behavior

Finally, we compared the behavioral effect of the 3D printed head cap system with manually built headmounts. To this end, we compared the running speed of rats and mice between subject implanted with the 3D printed and manually built headgears. Rats and mice were water deprived and had to collect water reward on a linear track or figure-eight circular maze. We observed no significant difference between the median running speed of the two rat groups (Kolmogorov-Smirnov test (KS-test); p=0.35) or the 95 percentiles (KS-test; p=0.95). We also performed the same test on

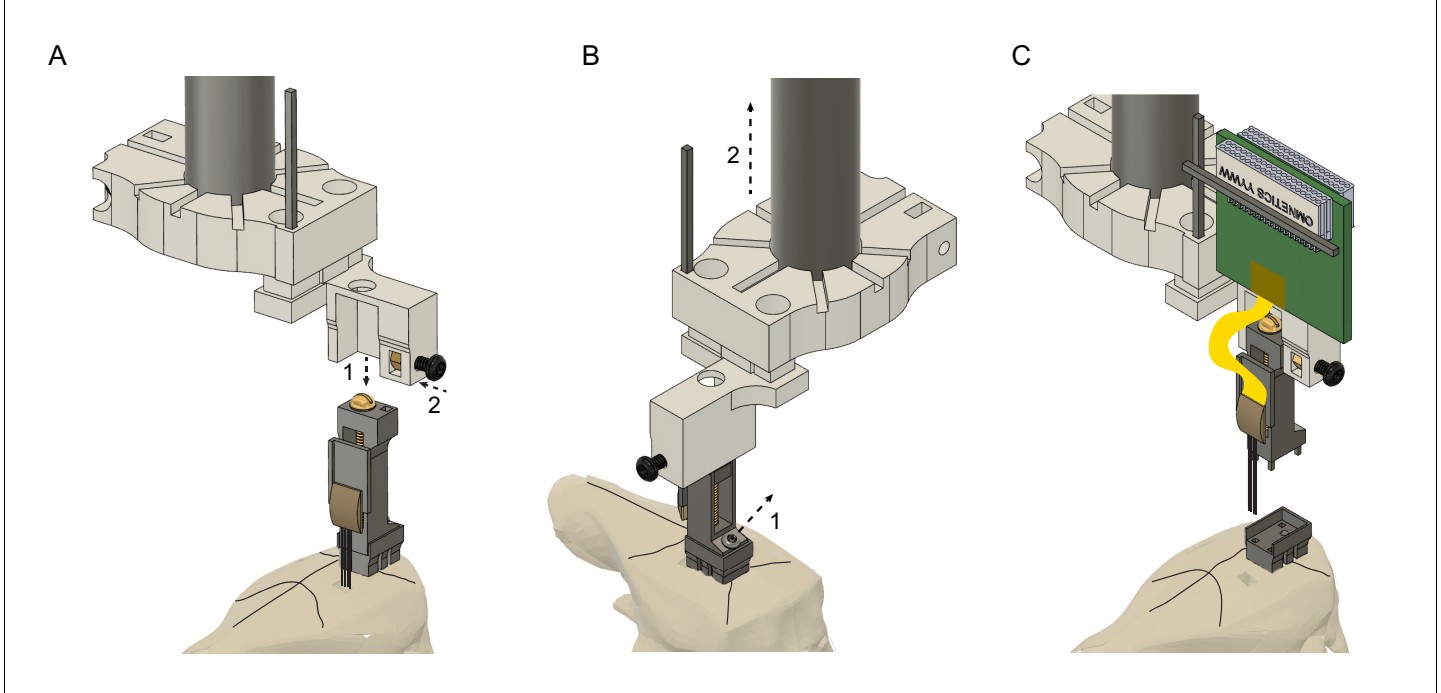

**Figure 4.** Probe recovery procedure. (**A**) The stereotaxic probe holder is attached to the microdrive (step 1) and is fixed with the black screw (step 2). Precise alignment is critical to avoid tissue damage and to prevent breaking the shanks when retracting the probe. (**B**) The microdrive is detached from the drive-base by removing the 000–120 steel screw (step 1) and moved upwards (step 2). Camera angle rotated 90°. (**C**) The drive with the attached probe after retracting it from the brain. The drive-base can be reused by cleaning it in chloroform or acetone.

The online version of this article includes the following video(s) for figure 4:

**Figure 4—video 1.** Silicon probe recovery from a mouse cap.

https://elifesciences.org/articles/65859#fig4video1

**Figure 4—video 2.** Silicon probe recovery from a rat cap.

https://elifesciences.org/articles/65859#fig4video2

mice and found a significant difference between the median running speed of the two groups (KS-test; p=0.045G) but no significant difference between the 95 percentile speeds (KS-test; p=0.24; *Figure 6H*).

## Discussion

We have developed a recoverable microdrive printed in stainless steel and a head cap system for chronic electrophysiological recordings in freely behaving rats and mice. The cap system allows for considerably faster and more standardized surgeries to be performed and faster post-surgical recovery of the animals. Importantly, recovery of the probe and head cap becomes an easy and routine procedure, allowing the same silicon probes to be used in multiple animals, offering substantial savings.

Our head caps are minimally invasive and do not require supportive bone screws, shortening surgery time and postoperative recovery. Except for the base, the entire headgear is reusable, making experiments performed on multiple animals less variable. For multiple surgeries (e.g., virus injection for optogenetic or pharmacogenetic experiments), implantation of the base during the first surgery provides fixed coordinates for a subsequent surgery. The head cap system is flexible, due to the large internal volume, and allows for multiple probe implants, optical fiber implants, and other optional components. In contrast, manually constructed cap systems are time-consuming to build, require extensive experience, and its construction may vary from animal to animal and across investigators even in the same laboratory. The main disadvantage of existing headgears is the limited success for probe recovery. Even after successful recovery of the recording probe, a new protective cap

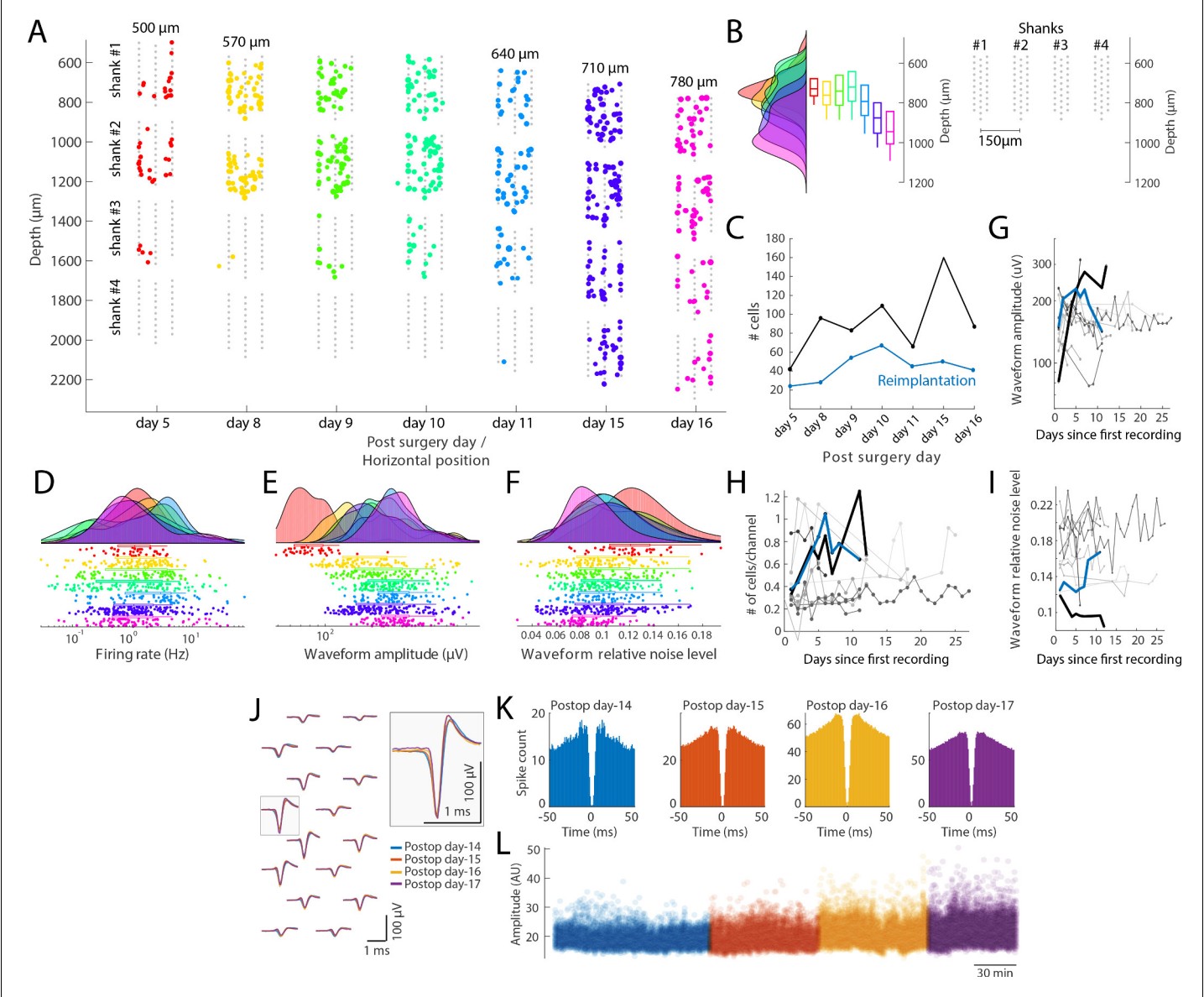

**Figure 5.** Single unit quantification. (**A**) Recordings from the prefrontal cortex at multiple depths across 12 days with a four-shank silicon probe in a mouse (128 channel probe from Diagnostic biochip; P128-5). Individual shanks are displayed as rows across days to better visualize the cells across days. The probe was moved in 70 μm steps to record from a new population of cells across days. Colored dots: position of single cells determined by spike amplitude trilateration. Grey dots: electrode sites. (**B**) Left: Histogram and boxplots of the distribution of recorded neurons as a function of cortical depth (μm) for each session shown in A. Each colored histogram and corresponding box-plot correspond to the same days shown in A. Right: Probe layout (shanks now shown in a horizontal layout) with the shanks shown with the depth for day 8–10; shanks are spaced by 150 μm. (**C**) Number of isolated single units across days after the first implantation (black) and after reimplantation of the probe (blue). (**D–F**) Firing rate (**D**), waveform amplitude (**E**; trough-to-peak) and relative noise level (**F**; waveform std divided by the waveform amplitude). (**G–I**) Comparison with other control mice and rats (n = 10 subjects), implanted with custom built drives (**Vandecasteele et al., 2012**), comparing waveform amplitude (**G**), number of cells/ recording site (**H**) and relative noise level (I; same definition as in panel F). Lines refer to different animal subjects. Thick black line: rat with the metal drive; thick blue line: rat with reimplanted silicon probe mounted on metal drive (panel A-F). Days relative to the first recording session from each animal. (**J–L**) Neuropixels probe recording, where the same putative interneuron was tracked across four days. (**J**) Average waveforms (bandpass filtered 300–10000 Hz) of a putative interneuron recorded on16 channels across 4 days (left). The average waveforms recorded at the site with the largest amplitude waveform is highlighted on the right (waveforms are color-coded by recording days). Autocorrelation histograms (**K**) and spike amplitudes (L; from Kilosort) for the same single unit, color-coded by recording day.

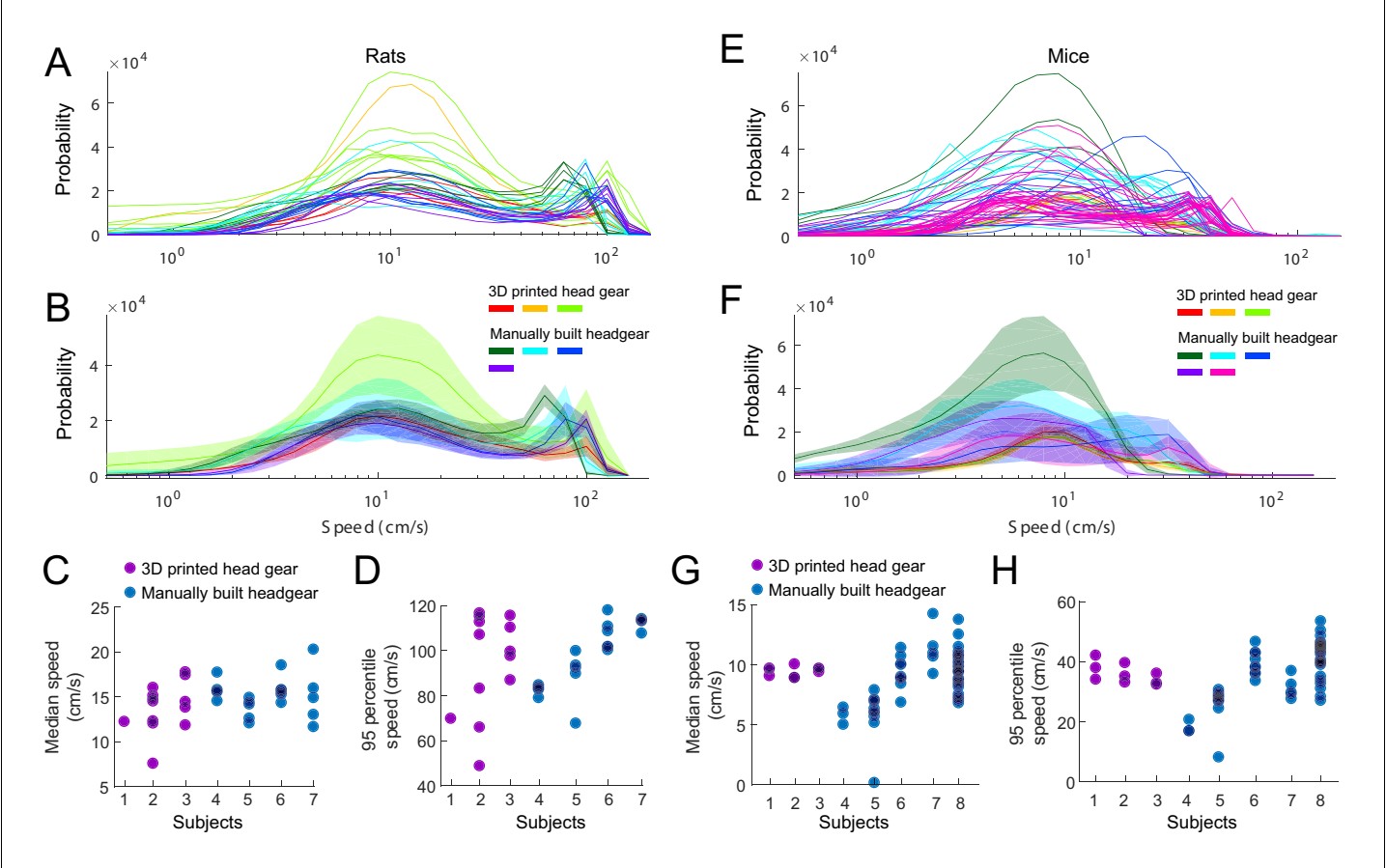

**Figure 6.** Effect of head gear type on running speed. Rats and mice implanted with the 3D printed head cap system and subjects with manually built headgear. (A–B) The distribution of running speeds within individual sessions (A) and within individual subjects (B). Three rats implanted with the 3D printed head gear (13 sessions), compared to four subjects with manually built headgear (22 sessions). (C–D) Median (C) and 95 percentiles (D) of the running speed, per sessions. (E–F) The distribution of running speeds within session (E) and average across sessions per subjects (F). Three mice implanted with the 3D printed head gear (nine sessions), compared to five control subjects with manually built headgear (54 sessions). (G–H) Median (G) and 95 percentiles (H) of the running speed, per sessions.

must be built from scratch in subsequent surgeries. In contrast, our modular cap system is prepared before surgery, decreasing the time the animal spends under anesthesia, reducing potential complications during and after surgery. Using this strategy, we were able to explant and implant the same silicon probe in >10 mice (*Senzai et al., 2019*).

The metal microdrive weighs 0.87 g with a footprint area of 15.5 mm$^2$, allowing the implantation of multiple probes in rats and even in mice. Because the entire headgear can be removed from the base with a screwdriver, recovery of the silicon probes is simple and highly successful. The drives are printed in stainless steel, with a stiffness prints approximately 100 times higher than that of plastic (Young's modulus of stainless steel: ~180 GPa vs. plastic: ~2 GPa). Steel drives provide higher stability, potentially better recording quality, and prevent potential wobbling while turning the screw to adjust the probe's position in the brain. Commercially available drives are typically built from plastic, are non-recoverable, and more expensive. Hand-made drives introduce variability across drives and experiments. In contrast, 3D steel printing provides high consistency across drives, reducing interexperimental variability.

Our system allows electrophysiologists to record the neuronal activity from multiple brain regions simultaneously in freely moving rodents. The large internal volume of the head cap and the small footprint of the metal microdrive enable researchers to perform more than one silicon probe implantation in freely moving mice and rats. Despite the highly successful recovery of silicon probes, the probe itself can deteriorate over time, limiting the number of reimplants (decreased signal-to-noise

over time, reduced number of high-quality single unit clusters). Appropriate cleaning procedures can extend the lifetime of silicon probes.

To facilitate wide use of the 3D printed designs, we share all necessary details of parts, fabrication process, and vendor source for easy replication by other laboratories (see Materials and methods). We offer several video tutorials, which describe the construction of the microdrive, the cap systems, the probe implantation, and the probe recovery. The CAD system allows different laboratories to customize both the drive and headgear according to their specific goals and needs.

# Materials and methods

**Key resources table**

| Reagent type (species) or resource | Designation | Source or reference | Identifiers | Additional information |
|---|---|---|---|---|
| Other | Recoverable drive (base) | 'This paper' – Github repository | Base_v7.step | |
| Other | Recoverable drive (drive) | 'This paper' – Github repository | drive_v7.step | |
| Other | Recoverable drive (arm) | 'This paper' – Github repository | arm_v7_50 um.step | |
| Other | 00–90 nut | McMaster | 92736A112 | |
| Other | 00–90 screw 1/2' | McMaster | 92482A235 | |
| Other | 00–120 screw 1/8' | McMaster | 96710A009 | |
| Other | Male header pin | DigiKey | SAM1067-40-ND | |
| Other | T1 and T2 screwdriver | McMaster | 52995A31 | |
| Other | 00–90 tap | McMaster | 2504A14 | |
| Other | 000–120 tap | McMaster | 2504A15 | |
| Other | stereotax_attachment_metal | 'This paper' – Github repository | stereotax_attachment_metal_v7.stl | |
| Other | 00–90 nut | McMaster | 92736A112 | |
| Other | 00–90 screw 1/4' | McMaster | 93701A005 | |
| Other | Male header pin | DigiKey | SAM1067-40-ND | |
| Other | drive_holder_metal | 'This paper' – Github repository | drive_holder_metal_v7.stl | |
| Other | 3 × 00–90 nut | McMaster | 92736A112 | |
| Other | 2 × 00–90 screw 1/4' | McMaster | 93701A005 | |
| Other | 00–90 screw 3/16' | McMaster | 93701A003 | |
| Other | 3D printed mouse cap (left wall) | 'This paper' – Github repository | left_wall_v12_L11.5 mm_W10.00mm.stl | |
| Other | 3D printed mouse cap (right wall) | 'This paper' – Github repository | right_wall_v12_L11.5 mm_W10.00mm.stl | |
| Other | 3D printed mouse cap (base) | 'This paper' – Github repository | mouse_base_v12_L11.5 mm_W10.00mm.stl | |
| Other | 3D printed mouse cap (cut out) | 'This paper' – Github repository | mouse_hat_copper Mesh_cutOut_v02.stl | |
| Other | 3 × 000–120 screw 1/8' | McMaster | 96710A001 | |
| Other | Male header pin | DigiKey | SAM1067-40-ND | |
| Other | Copper mesh | Dexmet | 3CU6-050FA | |
| Other | T1 screwdriver | McMaster | 52995A31 | |
| Other | 000–120 tap | McMaster | 2504A15 | |
| Other | 3D printed rat cap (left wall) | 'This paper' – Github repository | rat_cap_left_wall_v8.stl | |

*Continued on next page*

*Continued*

| Reagent type (species) or resource | Designation | Source or reference | Identifiers | Additional information |
|---|---|---|---|---|
| Other | 3D printed rat cap (right wall) | 'This paper' – Github repository | rat_cap_right_wall_v8.stl | |
| Other | 3D printed rat cap (base) | 'This paper' – Github repository | rat_cap_base_v8.stl | |
| Other | 3D printed rat cap (top) | 'This paper' – Github repository | rat_cap_top_v8.stl | |
| Other | 00–80 screw 1' | McMaster | 92196A060 | |
| Other | 00–80 brass insert | McMaster | 92395A109 | |
| Other | 2 × 00–80 screw 5/32' | McMaster | 92196A053 | |
| Other | Male header pin | DigiKey | SAM1067-40-ND | |
| Other | Copper tape | McMaster | 76555A724 | |
| Other | M2 × 0.4 thumb screw | McMaster | 99607A256 | |
| Other | M2 × 0.4 thin nut | McMaster | 93935A305 | |
| Other | 00–80 tap | McMaster | 2523A461 | |
| Other | 0.05' hex key | McMaster | 5497A22 | |
| Other | 3D printer | Formlabs | Form2 | |
| Other | Clear resin | Formlabs | RS-F2-GPCL-04 | |
| Other | Cotton swabs | Fisher Scientific | 19-062-616 | |
| Other | Kimwipes | Kimtech | 34120 | |
| Other | Gelfoam | Fisher Scientific | NC1861013 | |
| Other | Screwdriver | Amazon | B0058ECJIE | |
| Other | 000–120 screw 1/16' | Antrin Miniature Specialties | AMS120/1B-25 | |
| Other | Burrs for micro drill 0.7 mm | Fine Science Tools | 19008–07 | |
| Chemical compound, drug | C and B Metabond Base 10 ml | Parkell | P16-0116 | |
| Chemical compound, drug | C and B Gold Catalyst | Parkell | P16-0052 | |
| Chemical compound, drug | C and B Metabond Clear Powder | Parkell | P16-0121 | |
| Chemical compound, drug | Unifast Trad Powder Ivory | Pearson Dental | G05-1224 | |
| Chemical compound, drug | Unifast Trad Liquid | Pearson Dental | G05-1226 | |
| Chemical compound, drug | Unifast 1:2 Package A2 | Pearson Dental | G05-0037 | |
| Other | Dental LED Light | Aphrodite | AP-016B | |
| Chemical compound, drug | Cyanoacrylate | Loctite | 45208 | |
| Other | Ground/reference wire | Surplus Sales | (WHS) LW-12/36 | |
| Other | Ground/reference wire | Phoenix Wire Inc | 36744MHW - PTFE | |
| Chemical compound, drug | Ultrazyme Enzymatic Cleaner Tablets | Ultrazyme | B000LM0ZYS | |
| Other | Dieffenbach Vessel Clips Straight (rats) | Harvard Apparatus | ST2 72–8815 | |
| Other | Intan USB Eval board | Intan Technologies LLC | C3100 | |
| Other | Intan headstage | Intan Technologies LLC | C3324 and C3325 | |
| Other | Intan cable | Intan Technologies LLC | C3216 | |

## Microdrive assembly instructions

The base of the microdrive anchors the body of the microdrive via a tapped hole in the back (000–120 tap) and four rectangular holes inside the base ($0.5 \times 0.5$ mm$^2$). Thin walls around the base prevent cement flowing between the base and the body during surgery (*Figure 1A*). Glue a nut inside the arm (referred to as 'arm nut'; 00–90 brass nut) before attaching it to the body. The body has an opening in the top part of the back where a nut can fit inside ('top nut'; 00–90 brass nut). Insert the 'top nut' from the back, then insert the arm from the front and introduce a screw (00–90, 1/2', brass screw) through the 'top nut' and the 'arm nut'. Tighten the screw completely and release it a quarter-turn (or less). Fix the 'top nut' and the screw together using solder so the arm can be moved linearly relative to the body by turning this screw. Attach the body-arm complex to the base using a screw in the back (000–120, 1/8', stainless steel screw). Finally, insert a male header pin into the body and secure it using dental acrylic cement (Unifast Trad). This can be used as a soldering joint during surgery. Finally, attach the backend of the silicon probe to the arm using cyanoacrylate glue and solder the Omnetics connector (Omnetics Connector Corporation) of the probe to the male header pin of the drive holder. The fully assembled microdrive weighs 0.87 g (base: 0.225 g, arm with nut: 0.159 g, body with screw and metal bar: 0.486 g).

> *Assembly_instructions_microdrive_metal_v7.pdf* contains instructions with photographic documentation.
> *Figure 1—video 1* shows the assembly of the metal microdrive.
> *Figure 1—video 2* shows the attachment of a Neuropixels probe to metal microdrive.

## Implantation/recovery tool assembly instructions

Insert and glue one nut (00–90, brass nut) and a male header pin into the stereotactic attachment using cyanoacrylate glue. Insert a 00–90, 1/4' stainless steel screw into the nut. Tightening this screw will secure this piece to the electrode holder of the stereotactic arm (Model 1770, Kopf Instruments). The male header pin should be used as a temporary soldering joint for the Omnetics connector of the silicon probe. Insert and glue two nuts (00–90, brass nut) into the bottom of the drive holder and one nut (00–90, brass nut) into the body of the drive holder. Insert a 00–90, 3/16' stainless steel screw through this latter nut. This screw should be used to secure the top part of the body of the drive to the drive holder. Attach the stereotaxic attachment to the drive holder using 00–90 screws (00–90, 1/4', T2 screw).

> *Assembly_instructions_implantation_tool_metal_v7.pdf* contains instructions with pictures.

## Mouse cap assembly instructions

The base has a rectangular hole for a male header pin ($0.8 \times 0.8$ mm$^2$) for fixing the left and right walls temporarily during surgery (*Figure 2B*). This can help to open the cap using a fine pair of tweezers. The tip of the tweezer is squeezed between the rectangle and the walls. Pushing the tweezer against this rectangle readily opens the walls. The right wall has one tapped hole in the front and one in the lower part of the back (000–120 thread, 1.9 mm length). In addition, it has a hole in the upper part of the back (1 mm in diameter, 1.4 mm length). The left wall has one hole in the front and one in the lower side of the back (1 mm in diameter, 1.4 mm length) and a tapped hole in the upper side of the back (000–120 thread, 1.9 mm length). In addition, there are two rectangular holes in each wall ($0.8 \times 0.8$ mm$^2$) in which male header pins are glued with cyanoacrylate glue to serve as soldering points for the Omnetics connector and for the shielding copper mesh. To reduce weight, walls are perforated and covered with light copper mesh by gluing it with dental acrylic (Unifast Trad). The walls are closed using two screws in the back and one screw in the front (000–120, 1/8' stainless steel pan head torx screws).

> *Assembly_instructions_mouse_hat_10_39 mm_v11.pdf* file contains instructions with pictures.
> *Figure 2—video 1* shows the assembly of the mouse cap.

## Rat cap assembly instructions

The base has a hole for a brass screw-to-expand insert (00–80 thread size, 1/8' installed length) and serves to hold together the left and right walls. It has a rectangular protrusion in the back (3 × 1.5 × 1.67 mm³) to help opening of the cap using a fine tweezer. The right and left walls have a front hole (diameter 1.8 mm) in which a screw can be passed (00–80, 1' 18–8 stainless steel socket head screw) for fixing the walls to the metal insert of the base. In addition, there is a rail on each wall at the bottom part that grabs onto the base piece (1.2 mm height and 1 mm deep).

During surgery, the walls are kept open with the screw loosely tightened (*Figure 3B*, top part). After all the connectors are attached to the male header pins, the walls are closed, and the front screw is tightened. The right wall has a hole in the upper side of the back (1.8 mm, 2 mm length) and a tapped hole in the lower side of the back (00-80 thread, 2 mm length). The left wall has a hole in the lower side of the back (diameter 1.8 mm, 2 mm length) and a tapped hole in the upper side of the back (00-80 thread, 2 mm length). The walls are closed in the back using two screws (18-8 stainless steel socket head screw, diameter 0-80, 5/32" length). The left wall also has an insert in the upper part of the back side for a nut (18-8 stainless steel thin hex nut, M2.5 × 0.45 mm thread). This latter nut serves as a locking mechanism for the top cover. There are four rectangular through-holes in each wall (0.8 × 0.8 mm) in which male header pins are glued with epoxy (Araldite Standard Epoxy) and serve as soldering points. The locations of the holes can be modified according to user specifications to adapt different connector placements. To protect the implanted electrodes, the rat cap is covered by either self-adherent wrap (3M Coban) or the plastic top cover. The edge is extruded on the outer surface on top of the walls to provide extra surface for better adhesion. The plastic cover is attached to the walls using the front slide-in slot and the back screw (stainless steel flared-collar knurled-head thumb screw, M2 × 0.40 mm thread size, 4 mm long). To protect the neuronal signal from environmental electromagnetic interference noise, conductive copper coil electrical tape is glued to the walls by cyanoacrylate glue (copper tape: 1" wide, McMaster product number: 76555A724) and connected to the ground.

*Assembly_instructions_rat_cap_v8.pdf* file contains instructions with photographs.
*Figure 3—video 1* shows the assembly of the rat cap.

## 3D designing and printing parts

All parts were designed in Autodesk Fusion 360 (https://www.autodesk.com/products/fusion-360). We tested and printed cap designs on a Form two printer from Formlabs with 50 µm resolution using their standard resins. The metal microdrive prints were produced by Proto Labs (https://www.protolabs.com/services/3d-printing/direct-metal-laser-sintering). All designs are available from our GitHub repository https://github.com/buzsakilab/3d_print_designs (copy archived at swh:1:rev:a073716d89c32f13eb76a5ac5e7fa6f7fa11e18a; *Vöröslakos, 2021*).

Rat cap system: https://github.com/buzsakilab/3d_print_designs/tree/master/Rat_cap
Mouse cap system: https://github.com/buzsakilab/3d_print_designs/tree/master/Mouse_cap
Metal drive: https://github.com/buzsakilab/3d_print_designs/tree/master/Microdrives/Metal_recoverable

3D metal print submission procedure with Proto Labs.

1. Download the. step files from our GitHub.
2. Create a user account at Proto Labs (https://www.protolabs.com/) and upload the files.
    a. Add the following note to the drive body:' Use orientation as in quote 9301–742.'
    b. Choose Stainless Steel 316L material, high resolution and standard finish.

Proto Labs (Proto Labs, Inc, MN, USA) currently charges 250$ for a single print but substantial savings are available for larger print orders. Coordination of orders across laboratories therefore can reduce the price.

## Subjects

Rats (adult male Long-Evans, 250–450 g, 3–6 months old, n = 11) and mice (adult male C57BL/6JxFVB mice, 32–40 g, n = 6) were kept in a vivarium on a 12 hr light/dark cycle and were housed two per cage before surgery and individually after it. All experiments were approved by the

Institutional Animal Care and Use Committee at New York University Medical Center. Animals were handled daily and accommodated to the experimenter before the surgery and behavioral recording.

## Surgery instructions

The following instructions cover surgeries in both rats and mice, with differences highlighted. Prior to surgery, prepare the 3D printed cap, the microdrive(s), the implantation tool and attach a silicon probe to the microdrive (as described above).

We recommend measuring the impedance of the silicon probe before implantation using the RHD USB interface board from Intan (Intan Technologies LLC, CA, USA). Lower the probe into 0.9% saline and ground the saline to the recording preamplifier ground. Connect the probe to an Intan preamplifier headstage (RHD 32- or 64-channel recording headstages) and to the main Intan board to perform the impedance test measurement at 1 kHz frequency. This measurement provides a quick and rough estimate about the quality of the recording sites. If higher precision is required, users can perform impedance measurement with a nanoZ device following the manufacturers recommendation (nanoZ Impedance Tester, Plexon Inc, TX, USA).

### Prepare the stereotaxic apparatus and tools

1. Place the heating pad under the position of the ear bars.
2. Sterilize surgical instruments.
3. Weigh the animal subject.
4. Place all components in alcohol for disinfection.
5. Mice: prepare bupivacaine in an insulin syringe (0.4–0.8 ml/kg of a 0.25% solution).

### Surgery
#### Anesthesia and pre-incision preparations

1. The animal is anesthetized for 3 min (until after it loses its righting reflex) in an anesthesia-bucket with 2.5:1.5 (Anesthetic % to Airflow ratio).
2. Apply a local anesthetic to the tips of the ear bars before insertion (LMX-4 Lidocaine 4% topical cream). Fix the head with ear bars and attach the closed ventilation nosepiece. Once the animal is in the stereotactic apparatus, the level of anesthesia is lowered (1.2–2%).
3. Remove the hair above the planned surgery site using either Nair-hair remover or a hair trimmer.
4. Clean the hairless skin with the antiseptic solution and repeat the process two more times (Povidone-Iodine – 10% topical solution). Apply the antiseptic solution with Kimtech wipes using anterior to posterior swipes. The last swipe must be done in one stroke to minimize infections. Between each swipe with the antiseptic solution, the skin is cleaned by 70% alcohol applied with the same technique.

#### Incision and skull cleaning

1. Inject bupivacaine (0.4–0.8 ml/kg of a 0.25% solution) subcutaneously along the scalp midline for local anesthesia. Make one injection site and distribute the anesthetics along the midline.
2. Make a median incision from the level of the eyes to the back of the skull (neck).
3. Separate the skin from the skull, pull the skin sidewise and attach four bulldog clips to create a rectangular shape opening. The bulldog clips should be attached to the subcutaneous soft tissue, not the skin.
4. Scrape the skull with a scalpel and remove the periosteum from the top flat surface of the skull. This is necessary to achieve a strong bond with the 3D printed base.
5. Clean the skull surface with saline and vacuum suction.
6. Clean the skull with hydrogen peroxide and rinse it with saline. The hydrogen peroxide is applied with cotton swabs (about 5 s) and rinsed quickly thereafter thoroughly with saline. Avoid touching the skin and muscle with the solution.
7. Cauterize any bleedings along the skull and exposed skin.

## Attaching the base to the skull

1. Prepare the Metabond on ice. Mix four drops of base with 1 drop of catalyzer.
2. Paint, using a brush, the whole surface of the cleaned and dried skull and let it dry.
3. Mix a new solution of Metabond with powder: four drops of base, one drop of catalyzer and 2 scoops of powder and apply a second layer of Metabond paint to the skull surface. Paint also along the edge of the skull surface.
4. Paint the bottom surface of the 3D printed base with Metabond and align it above the skull and attach it to the skull before it solidifies.
5. Paint with Metabond along the inner contact line between the hat base and the skull and create a sealed area inside the hat.
6. Gently hold the hat base in place (for about for 60 s) until it stays attached to the skull using your fingers. Let the Metabond cure before proceeding to the next steps.

## Craniotomy marking and screw placement

1. Align Bregma and Lambda in the same horizontal plane. Determine the position of Bregma using stereotactic coordinates with a fine needle attached to the stereotactic arm.
2. Calculate the relative positions of the probe incision points.
3. Mark the positions of the planned craniotomies with a scalpel (gently make two orthogonal lines crossing at the planned incision points with the scalpel) and a pen (fill the scalpel-drawn lines with the pen).
4. Mark the position of the reference and ground screws with the scalpel/pen.
5. Remove the stereotactic arm.
6. Drill holes for ground and reference screws in the skull above the cerebellum with a high-speed drill. If bleeding occurs, rinse it with saline and vacuum suction until the bleeding stops.
7. Insert the ground and reference screws in. Begin with a slight counterclockwise turn. For mice, allow a margin of about 0.5 mm. In rats, drive the screws tight. Alternatively, 125 µm stainless steel wires can be used for reference and ground, instead of screws.

## Craniotomy

1. Perform the craniotomy with a high-speed drill (drill bit size depends on the goal). Rinse it with saline and vacuum suction to ensure visibility while drilling.
2. Clean around the craniotomy with the drill or a scraping/sharp scooping tool.
3. Remove the dura with a hook-shaped needle at the planned incision site for probe insertion: bend the tip of the 30G needle to form a small hook (gently tap the tip of the needle into a hard surface to form the hook). Lift the dura with the hook and cut with a pointed scalpel (size 11). Avoid damaging blood vessels.
4. Apply saline and Gelfoam to the craniotomy to maintain a wet brain surface.

## Probe implantation

1. Place the silicon probe in the implantation tool on the stereotactic arm and position it according to the specified surface coordinates.
2. Lower the silicon probe to the brain surface at the marked coordinates.
3. Insert the probe to the desired target depth in the brain.
4. Fix the base of the microdrive to the skull and hat-base with regular grip cement.
5. Apply silicone to the craniotomy, let the silicone run along the shanks and seal the craniotomy completely. This protects the brain and limits bleedings and blood coagulation. Alternatively, apply a mixture of paraffin oil/wax to the craniotomy with a needle and heat it using the tip of a soldering iron.
6. Solder the reference and ground wires to the corresponding sites on the Omnetics connector.
7. Attach the cap sidewalls to the base.
8. Cover the top with the lid or Coban tape.
9. Turn off the anesthesia and release the animal from the stereotactic setup.

## Post-operative care

1. Weigh the animal after surgery to determine the weight of the added headgear.

2. Place the animal back in a home cage. The cage should be placed on a heating pad during the first night.
3. Inject Buprenex subcutaneously after 20 min (0.05–0.1 mg/kg).

### General notes

- Apply mineral oil to the eyes of the animal at regular intervals.

- To keep the animal properly hydrated during the postoperative days, provide an aqua-gel and a small container with water. Provide regular rodent pills.

## Additional implantation information

Rats and mice were implanted with silicon probes to record local field potential and spikes from the CA1 pyramidal layer in rats and from the prelimbic cortex from mice. Silicon probes (NeuroNexus, Ann-Arbor, MI and Cambridge Neurotech, Cambridge, UK) were implanted in the dorsal hippocampus (rats: antero-posterior (AP) −3.5 mm from Bregma and 2.5 mm from the midline along the medial-lateral axis (ML); mice: antero-posterior (AP) +1.75 mm from Bregma and 0.75 mm from the midline, 10 degree relative to the sagittal axis). The probes were mounted on the recoverable metal microdrive and previous design iterations made in plastic (unpublished work; stl files and instructions are available at our GitHub repository https://github.com/buzsakilab/3d_print_designs/tree/master/Microdrives/Plastic_recoverable), allowing for precise vertical movement after implantation and implanted by attaching the base of the micro-drives to the skull with dental cement (*Supplementary file 1*). The small footprint of the metal microdrive enables researchers to perform more than one silicon probe implantation in freely moving mice. For this purpose, larger mice (>35 g) were selected.

After the post-surgical recovery, we moved the probes gradually in 50 μm to 150 μm steps until the tips reached the pyramidal layer of the CA1 region of the hippocampus. The pyramidal layer of the CA1 region was identified by physiological markers: increased unit activity and the presence of ripple oscillations (*Mizuseki et al., 2011*). In mice, the probe was implanted 500 μm below the surface of the brain and recordings were performed each day. The probe was moved 70 μm after each recording day. Data was collected daily. The implanted animals were single housed, and they do not carry the headstage while in the vivarium. During recordings, the headstage is attached and a counterbalanced pulley system ensures that the animal is not carrying the extra weight of the headstage and cable.

## Electrophysiology data

Electrophysiological recordings were amplified using an Intan recording system: RHD2000 interface board with Intan 32 and 64 channel preamplifiers sampled at 20 kHz (Intan Technologies). All data is available from https://buzsakilab.com/wp/database/ (*Petersen et al., 2020a*).

## Behavioral data

Rats were trained to run on a 2.3 m linear track, or a 120 cm diameter circular track with a diagonal path allowing the animals to run in a figure-eight pattern (n = 3 rats implanted with the 3D printed head gear, n = 13 sessions, and n = 4 control subjects with manually built headgear, n = 22 sessions). In both behavioral paradigms, rats were water deprived and had to collect water reward (~0.02 ml).

Mice were either trained to run on a 1.1 m linear track (n = 3 mice implanted with the 3D printed head gear, n = 9 sessions), or to run on a 80 cm diameter circular maze with a diagonal path allowing the animals to run in a figure-eight pattern (same layout as the rats but a smaller maze, n = 5 control subjects with manually built headgear, n = 54 sessions). In all behavioral paradigms, mice were water deprived and had to collect water reward (~0.02 ml).

The linear tracks had 'reward areas' on each end where water reward was delivered via a custom-built infrared-beam triggered system. Animals only received water reward for trials in which they travelled from one reward site to the other. On the circular maze the animals ran along the central

arm after which they ran along the outer circle in a alternation fashion. Water reward was delivered in the reward area on correct trials.

The behavior of the animals was recorded using the Optitrack camera system (NaturalPoint, Inc, OR, USA).

### Spike sorting and data processing

Spike sorting was performed semi-automatically with KiloSort (*Pachitariu, 2016*) https://github.com/cortex-lab/KiloSort, using our pipeline KilosortWrapper (a wrapper for KiloSort, https://github.com/petersenpeter/KilosortWrapper) (*Petersen et al., 2020b*), followed by manual curation using the software Phy (https://github.com/kwikteam/phy) and our own designed plugins for phy (https://github.com/petersenpeter/phy-plugins). Finally, we processed the manually curated spike sorted data with CellExplorer (*Petersen and Buzsáki, 2020*) and performed further analysis with custom Matlab code.

## Acknowledgements

We thank Manuel Valero, Antonio Fernandez Ruiz, and Kathryn McClain for useful comments on the manuscript. We also thank Viktor Varga, and Kathryn McClain for behavioral data and Thomas Hainmueller for intraoperative photographs. Supported by U19 NS107616, U19 NS104590, R01 MH122391, and The Lundbeck Foundation.

## Additional information

### Funding

| Funder | Grant reference number | Author |
| --- | --- | --- |
| National Institutes of Health | U19 NS107616 | György Buzsáki |
| National Institutes of Health | U19 NS104590 | György Buzsáki |
| National Institutes of Health | R01 MH122391 | György Buzsáki |
| Lundbeckfonden | R271-2017-1687 | Peter C Petersen |
| Danish Council for Independent Research, Medical Sciences | DFF-5053-00279 | Peter C Petersen |

The funders had no role in study design, data collection and interpretation, or the decision to submit the work for publication.

### Author contributions

Mihály Vöröslakos, Conceptualization, Software, Investigation, Methodology, Writing - original draft; Peter C Petersen, Conceptualization, Software, Funding acquisition, Investigation, Methodology, Writing - original draft; Balázs Vöröslakos, Conceptualization, Methodology; György Buzsáki, Conceptualization, Supervision, Funding acquisition, Writing - original draft

### Author ORCIDs

Mihály Vöröslakos ORCID https://orcid.org/0000-0002-1022-1355
Peter C Petersen ORCID https://orcid.org/0000-0002-2092-4791
György Buzsáki ORCID https://orcid.org/0000-0002-3100-4800

### Ethics

Animal experimentation: All experiments were approved by the Institutional Animal Care and Use Committee at New York University Medical Center (protocol number: IA15-01466).

### Decision letter and Author response

Decision letter https://doi.org/10.7554/eLife.65859.sa1

Author response https://doi.org/10.7554/eLife.65859.sa2

## Additional files

### Supplementary files

• Supplementary file 1. Summary of experiments using recoverable microdrive and cap system in rodents. All animals were implanted with either mouse (M) or rat (R) cap system. The cap system was first tested in a mouse with wire electrodes (M_01) and in a rat without electrophysiology (R_01). After the evaluation of the cap system, we have implanted silicon probes attached to recoverable, plastic microdrives (Plastic recov refers to this microdrive) or flexible probes cemented in place and tested the cap system with electrophysiology in freely moving mice (M_02 – M_04) and in freely moving rats (R_02 – R_06). We used silicon probes from NeuroNexus (Buzsaki-32 NN) and Cambridge Neurotech (ASSY-156 E1 CN). Finally, we evaluated the recoverable, metal microdrive using a Diagnostic Biochips probe (128–5 DB) in a freely moving mouse (M_05) and a Neuropixels probe in a freely moving rat (R_09). All attempted probe recovery was successful except in R_02A (two shanks broke during the acute recordings). The same ASSY-156 E1 (CN) probe was used in R_03 and R_04, but after successful recovery from R_04 the impedance of the contact sites were too high to reimplant this device. A new ASSY-156 E1 probe was used in R_05.

• Transparent reporting form

### Data availability

All documentations for parts and device fabrication are included in the manuscript and supporting files, including video recordings. The same information is made public via GitHub (https://github.com/buzsakilab/3d_print_designs/tree/master/Microdrives/Metal_recoverable; copy archived at https://archive.softwareheritage.org/swh:1:rev:a073716d89c32f13eb76a5ac5e7fa6f7fa11e18a) and the associated website: https://buzsakilab.github.io/3d_print_designs/. Data from example electrophysiological recordings are available here (https://buzsakilab.com/wp/projects/entry/65723/).

The following dataset was generated:

| Author(s) | Year | Dataset title | Dataset URL | Database and Identifier |
|---|---|---|---|---|
| Vöröslakos MI, Petersen PC, Buzsáki Gr | 2021 | Metal microdrive and head cap system for silicon probe recovery in freely moving rodent | https://buzsakilab.com/wp/projects/entry/65723/ | Buzsaki lab databank, 65723 |

The following previously published dataset was used:

| Author(s) | Year | Dataset title | Dataset URL | Database and Identifier |
|---|---|---|---|---|
| Petersen PC, Buzsáki Gr | 2020 | Theta rhythm perturbation by focal cooling of the septal pacemaker in awake rats | https://buzsakilab.com/wp/projects/entry/4919/ | Buzsaki lab databank, 4919 |

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
