## [Decision Letter]

**Acceptance summary:**

The authors describe a customizable and versatile microdrive and head cap system for silicon probe recordings in freely moving rodents. The added value in this work beyond previous designs is: a) a carefully designed solution to facilitate probe recovery, thus reducing experimental costs and favoring reproducibility; b) flexibility to accommodate several microdrives and additional instrumentation; c) open access design and documentation to favor customization and dissemination. The design is quite simple and versatile, and the authors provide detailed instructions. This will be a very useful resource for many readers who are interested in performing multi-site silicon probe recordings of large ensembles of neurons in freely behaving rodents. The method will appeal to a broad range of systems neuroscientists who seek to understand neurophysiological mechanisms underlying cognition and behavior.

**Decision letter after peer review:**

Thank you for submitting your article "Metal microdrive and head cap system for silicon probe recovery in freely moving rodent" for consideration by *eLife*. Your article has been reviewed by 3 peer reviewers, and the evaluation has been overseen by Laura Colgin as the Senior Editor. The following individuals involved in review of your submission have agreed to reveal their identity: Liset M de la Prida (Reviewer #1); Michael Okun (Reviewer #2), and Ashley L Juavinett (Reviewer #3).

The reviewers have discussed their reviews with one another, and the Reviewing Editor has drafted this to help you prepare a revised submission. We have also prepared public reviews of your work, which are designed to transform your unrefereed author manuscript into a publicly accessible refereed preprint (read more about this in the "Posting public reviews" section below).

Essential revisions:

Reviewers were enthusiastic about the potential impact of this work. However, some concerns were raised regarding transparency, specific quantitative support of usability measurements, and clearer descriptions regarding the novelty of this resource compared to other previously published resources. Showing long-term recordings is recommended, but it is expected that the authors already have these recordings on hand. That is, reviewers believe that no new data are required, simply an inclusion and analysis of existing data. Authors are also requested to be more specific and transparent about which probes work with this resource. Specific details can be found in the three independent reviews, which were remarkably consistent and are included in their entirety below. Reviewers would like to see authors addressing all of their comments.

*Reviewer #1 (Recommendations for the authors):*

Comments and suggestions (the order is not informative)

1. The report reads very descriptive. While authors declare having used n = 7 rats and n=5 mice in methods, this is not exploited quantitatively in the paper. The ms will benefit from including some assessment to support ease of use (e.g. behavior in implanted versus not implanted animals), stability of recordings over days, weeks or months; or any other relevant supportive example.

2. Page 4, line 100 and table 1: What travel distance this refers to? The effective travel distance should be better explained. Once the probe is mounted in the shuttle, the travel distance is limited by the probe length and tip location from the microdrive base. Unexperienced users may misunderstood this important issue.

3. Table 1: What are inclusion criteria? Are these microdrives designed for tetrodes or for silicon probes; mice only? Korshunov 2006 is for single wires in a wide range of species and I fail to see how it could be repurposed for silicon probes. If the goal of the table is to summarize existing solutions, then it may fall short for including other microdrive designs. I would recommend focusing. Is Voroslakos et al., referring to this study ? Please, clarify.

4. Figure 1D: the stereotaxic microdrive holder is very similar to that in Chung et al., (Figure 4). Is that the case? What are the slots in the attachment for?

5. Figure 2F-H and Figure 3D: In general, data is very descriptive and poorly relate with the potential added value of the system. I would consider providing some quantitative assessment to prove stability, or reusability, or consistency of some LFP feature between animals. What metrics is shown in Figure 2H and why is this useful to illustrate the added value of the microdrive? Same for Figure 2F.

6. Introduction, line 49: authors make a good case for the emergence of novel integrated solutions such as uLED and microfluidic probes which benefit from the use of versatile headgear designs. However, both Wu et al. and Kim et al. seem to report on uLEDs only. Consider Altuna et al., Lab on Chip 2013 https://doi.org/10.1039/C3LC41364K of any other appropriate reference for integrated fluidic multi-site probes, or just avoid the mention to microfluidity.

7. Introduction, lines 52-54: flexDrive, shuttledrive, etc. all of them are mostly dedicated to carry tetrodes. There are fewer designs for silicon probes, which may support the need for this paper, but authors avoid discussing the potential overlapping and added value between existing solutions and their own design. I feel the introduction will benefit from addressing this more sharply.

8. Videos: Only two videos are provided to illustrate the head-cap systems of rats and mice. While documentation provided is useful, many relevant parts of the paper will strongly benefit from providing video support (e.g. microdrive assembly, implantation, recover).

9. Methods, line 352: stl files are provided via a Github. It would be useful to upload all of them together as supplementary material of the paper itself. Also, please, consider adding specific links to the different files repository in the Methods section.

10. Methods, line 369: authors recommend measuring probe impedance. This is not particularly easy with silicon probes. Please, add equipment information.

*Reviewer #2 (Recommendations for the authors):*

This manuscript provides an updated guide on the procedures for performing chronic recordings with silicon probes in mice and rats in the lab of the senior author, who is one of the leaders in the use of this experimental method. The new set of procedures relies on metal and plastic 3D printed parts, and represents a major improvement over the older methodology (i.e. Vandecasteele et al. 2012).

The manuscript is clearly written and the technical instructions (in the Methods section) seem rather detailed. The main concerns I had are as follows.

1. The present design is an improvement over Chung et al., (the most similar previously published explantable microdrive design, as far as I am aware) in terms of the footprint and travel distance. However, a main disadvantage of the system in its present form is that (apparently) it does not support Neuropixels probes. While such probes might not be suitable for some uses (e.g. to record from large populations in dorsal hippocampus), Neuropixels probes are of considerable interest to many labs.

2. The total weight of the mouse implant seems quite high (together with the headstage, I estimate it is >= 4gr). Could the authors provide the exact value, and describe whether this has any impact on the way the animal moves? Also, the authors should describe how the animals are housed (e.g. do they carry the headstage even when not being recorded). The authors say that a mouse can be implanted with more than one microdrive. The authors should clarify whether they actually have an experience with such implants, or is this just a suggestion based on their educated estimate?

3. There is no information in the Results section on the number of implants performed, the duration the animals were implanted, the quality of the recordings obtained, number of successes or failures. The figures merely provide examples of one successful recording in a mouse and in a rat. All these details should be provided, along with details of how many probes were reused and how many times (a brief mention of one case, lines 252-253 and 359-360, is not sufficient).

4. In Figure 2, spike waveforms are classified as pyramidal, wide or narrow interneurons. I did not find any description of how this classification was performed.

5. Also in Figure 2, refractory period violations are reported in percent (permille in fact). First, it is not clear how refractory period was defined. Second, such quantification is incorrect in principle: we use refractory period violations to infer the rate of false positives. Yet the relationship between fraction of ISI violations and false positive rate depends on the firing rate of the neuron. For example, 0.1% of ISI violations is quite good for a unit spiking at 10 spikes/s, is so so for a unit spiking at 1 spike/s, and is very bad if the firing rate is 0.1 spike/s (see Hill et al. JNeurosci. 2011 for derivation). Alternatively, the authors can follow an approach described in an old paper by the same lab (Harris et al., JNeuropsysiol. 2000), quantifying the violations in spike autocorrelogram relative to its asymptotic height.

6. Line 477: the authors write that the probes were mounted on a plastic microdrive. This seems to contradict the key claim of the manuscript (namely that the microdrives were from stainless steel).

7. I believe that the work of Luo and Bondy et al., (*eLife* 2020) and should be references and compared to.

*Reviewer #3 (Recommendations for the authors):*

First, I'd like to applaud the authors for the development of a very clever device and for their clear description of how to use it. With the inclusion of more details around the compatibility of this device with specific probes, more support for specific claims in the manuscript, and more transparency about the success of probe recovery, this manuscript will inevitably serve as an important resource for many researchers.

I'll leave one thought for consideration here before diving into specifics: the organization of this manuscript was a bit unclear to me. I think it makes sense to have a "Results section" which gives a high level description of the procedure and your design choices, as well as a "methods section" which outlines the protocol, but this needs to be clear. This could be solved by a line at the end of the introduction that says something like "Here we'll describe the results we obtained and a high level summary …. Readers can find detailed instructions in the methods as well as in the attached files…". I would also defer to the *eLife* editors for how they would like to handle this organization.

The introduction of the paper is very well written, but towards the end there are several unsubstantiated claims. Specifically, the idea that this design reduces surgery time substantially. Can you be more specific, or back this up with timelines from other designs? On line 203 there is a reference to another paper after this claim, but this could be made more clear in the introduction. Which aspect of this procedure is quicker than other procedures? As I'll come to later in this review, there is also a claim about the recovery being "reliable" – knowing exactly how reliable, given your experience, would be extremely useful.

Overall, I have two suggestions that would greatly improve this manuscript.

First, more neural data should be shown. In Figure 2 E-H and Figure 3, some neural data recorded using this device is shown, but this is not nearly enough for users to assess the usability of these probes. The mouse data is particularly sparse, and it is very difficult to make much out of Figure 3D without seeing isolated units. If 7 rats and 5 mice were recorded, more of this data should be shown. Specifically, users may be interested in seeing the stability of units over time, as well as the SNR levels on various days of recording. The methods there was daily recording – showing some of this data would be useful. How long has one of these devices been successfully used to record activity? These types of details are essential to ensuring that your device enables quality data collection.

Secondly, there should be a table detailing each procedure done with these devices, including the type of animal (mouse/rat), age (if available), sex, success of recording (and for how many days/weeks), and success of the probe recovery (beyond saying it is "highly successful", line 257). This comprehensive overview of exactly how reliable your device is will be very useful to readers.

Several aspects of the manuscript could be clarified.

1. The "footprint" of the device is given multiple times, but a height and width would also be useful.

2. There is a clear trade-off between the minimal wear and increased reusability of metal drives with the weight of these drives, and that should be acknowledged. Would it be possible to create such a drive with a sturdy but lighter plastic, and if not, why?

3. Line 105 says the microdrive weighs 0.87 g – is this without any materials to attach it to the skull? Similarly, the overall weight of the entire assembly should be given.

4. It is unclear what material the stereotaxic attachment is made of.

5. There doesn't seem to be any mention of how to actually attach the probe to the arm/shuttle. Is it glued? Which probes were used should be clear in the Results section (I see it is eventually mentioned in the methods). Relatedly, it should be clear which types of probes were tested with this microdrive, and which probes you would recommend using with it. Specifically, will this microdrive and assembly work with Neuropixels 1.0 probes? Being clear about which probe was used is especially important for the multi-probe implantation – presumably this will not work with probes with large PCB boards and/or headstages. Also, are multi-probe implantations possible in mice using your microdrives and assembly?

6. It is unclear how the headstage is affixed in either the rat or mouse assembly.

7. For probe recovery, it's important to note that distilled water will not be recommended for all probes. For example, neuropixels have very clear restrictions on what you should use with them. I would advise the reader accordingly. Tergazyme may be useful here as well.

8. How is grounding handled in these devices? There are multiple mentions of a skull screw used to affix the protective assembly, but in most designs, a skull screw is there to serve as a reference and/or grounding. (Sidenote: It is not clear to me why a ground screw is so bad for the animal, as is emphasized multiple times in the manuscript. We are also putting a large open whole in the skull…) Is there a ground wire in these devices? Similarly, is the copper mesh inside electrically connected to the probe, or is it kept isolated?

9. Line 476 in methods is very unclear and mentions a plastic microdrive: "The probes ere mounted on a plastic recoverable microdrive to allow precise vertical movement after implantation (github.com/YoonGroupUmich/Microdrive) and implanted by attaching the base of the micro-drives to the skull with dental cement." Is this the same microdrive mentioned in the main manuscript? In general, it is unclear how the "Additional implantation information" relates to the main methods and seems that this information should be integrated into the other methods sections.

10. There is no mention of where to download the design files.

11. Data is provided for only 2 animals (one mouse, one rat) and 2 sessions – could more data be made available?

12. Code is provided for sorting (Kilosort Wrapper and phy plugin).

[Editors' note: further revisions were suggested prior to acceptance, as described below.]

Thank you for resubmitting your work entitled "Metal microdrive and head cap system for silicon probe recovery in freely moving rodent" for further consideration by *eLife*. Your revised article has been evaluated by Laura Colgin (Senior Editor) and a Reviewing Editor.

The manuscript has been improved but there are some remaining issues that need to be addressed, as outlined in the individual reviews below:

*Reviewer #1 (Recommendations for the authors):*

Authors have addressed all comments. The revised version is substantially improved. Videos are superb useful as well as the accompanying information.

*Reviewer #2 (Recommendations for the authors):*

I would like to thank the authors for carefully and thoroughly addressing the concerns and comments that I have raised. I believe that the revised version (including the addition of the videos illustrating the key procedures) is significantly improved.

I would like to encourage the authors to incorporate into the final version of the paper all the relevant technical details from the rebuttal. For example, in their response the authors mention using a pulley to counterbalance the headstage (yet this seems not to be mentioned in the manuscript), and similarly with preferentially using large (>35gr) mice.

*Reviewer #3 (Recommendations for the authors):*

This manuscript has improved significantly since the initial submission with numerous additional figures, tables, and methodological details that address the initial concerns. The reviewers have added multiple useful insights to the manuscript, including the footprint of the device, more clear justification for the use of stainless steel, the material of the implantation tool. The additional videos explaining how to assemble the headgear are very well done and clear. These details will undoubtedly help readers who wish to implement this headgear system in their own labs. The addition of more quantification of the neural data is also very appreciated. The authors have added additional analysis for Figure 2, showing single units for the mouse recording (though it would be useful to have a similar analysis in Figure 3 for the rat data).

There are several points that should be addressed within the revised manuscript:

The new Figure 5 illustrates the use of two different recording devices in mice. Figure 5A demonstrates the units recorded over time on each shank as the probe is lowered each day. However, it is unclear to me how Figure 5B relates to 5A – in 5B it looks as if all four shanks are at the same depth, whereas in 5A they are at different depths. Ultimately, it seems like some sort of integration representation of these two panels where viewers can appreciate the location of single units on each shank over days would be the most useful. On this same figure, the axis labels for 5F and 5I are a bit misleading, because this is not about the noise level of the recording, but about the waveform. I'd suggest changing the figure and the wording in the text to "Waveform relative noise level" so that readers do not confuse this with overall signal to noise in the recording. The axis on 5G could be readjusted so that readers can appreciate the data. In the figure caption, I'd suggest spelling out "ACGs" for readers unaccustomed to that shorthand. Presumably these ACGs are from the unit in the box? If so, I'd clarify in the caption.

Related to Figure 5, the corresponding text says, "suggesting either that the distance between the electrode sites and neuron bodies decreased or that large size neurons were recorded," but it is important to note that the probe was being moved into different brain areas over these days of recording. The text should be modified to clarify this – the clear difference from day 5 to the other days is almost definitely explained by the movement to deeper brain structures. This paragraph should also note the clear decrease in the # of units with the reimplanted probe, as well as the clear increase in the noise level in the reimplanted probe.

The authors have also added behavioral data, shedding light on the ability of animals to move with this headgear, however some clarity around these behavioral findings is needed. On line 219 it reads, "The 3D printed head cap system is comparable in weight to manually built headmounts" which is an abrupt and unclear transition to the paragraph about the impact of the headgear on behavior – reading between the lines, I think the authors are saying, "… however, we wanted to verify still that our headgear would not impede the animal's behavior." It is also unclear from this paragraph how this behavior was measured. The methods section describes mice and rats running on a track – were they headfixed? It is also unclear how the water reward is relevant here – did animals need to collect a water reward in order for the track to continue moving? Also, the mice with 3D printed headgear ran on a different track than the mice with manually printed headgear – some mention of this in the main text is warranted for transparency in interpreting the behavioral results shown. The wording in the methods is also unclear – is the track circular or a figure eight? Finally, contextualizing the results found here in terms of typical speeds on such a treadmill (e.g., in the discussion, see the point below) would be very useful to readers.

In this same paragraph, the authors state they found a "small significant difference," but this wording is misleading. A statistical test result is or is not significant, and the authors should remove the word small. Readers can determine whether or not the absolute value of the p-value is informative. There is a typo in this section also: "We also performed the same test on mice subjects and found a small significant difference between the median running speed of the two *rat* groups (KS-test; p = 0.045) but no significant different between the 95 percentile speeds (KS-test; p = 0.24)." I would also recommend that the authors write out the full name of the KS test, at least on the first mention.

Line 139 says, "High-quality electrophysiological signals can be collected from freely moving mice for weeks and months (Figure 2E and F)" however it is unclear from this figure whether this data was indeed recorded over weeks and months. If this is a claim the authors cannot substantiate with data, it should be removed. Figure 5 does indeed show "weeks" with the headgear (showing data out to day 16 for the 4-shank probe and 17 for neuropixels), so it seems like an applicable figure to reference here, without the mention of "months." Similar changes should be made for the reference to the rat recordings and Figure 3 – it's unclear based on the figure caption how long after implant these extracellular traces were obtained.

Finally, although these are significant additions to the main portion of the text, there is no mention of either the single-unit figure or the behavioral results in the discussion. A contextualization of these data in terms of the headgear's usability as well as a comparison of these findings to other recoverable drives would be useful and warrant discussion.

---

## [Author Response]

Reviewer #1 (Recommendations for the authors):Comments and suggestions (the order is not informative)1. The report reads very descriptive. While authors declare having used n = 7 rats and n=5 mice in methods, this is not exploited quantitatively in the paper. The ms will benefit from including some assessment to support ease of use (e.g. behavior in implanted versus not implanted animals), stability of recordings over days, weeks or months; or any other relevant supportive example.

We added a table to the manuscript to better describe what kind of cap and microdrive system was used in each animal (Supplementary File 1). Our developmental process was the following: first, we tested the ease of use and longevity of the cap systems in a chronic rat (R_01, n = 90 days) and a chronic mouse (M_01, n = 60 days). After this initial test, we implanted flexible probes with the cap system to test longevity of electrophysiology recordings using our cap system both in a mouse and a rat (M_02, 32-channel flexible probe, n=27 days and R_02, 64-channel flexible probe, n = 76 days). Finally, we tested the cap systems with our recoverable, plastic microdrive system which we have been using for the last two years in our laboratory. We used 32 and 64-channel silicon probes, Neuropixels probes as well as tungsten wire electrodes in mice and rats (for details see Supplementary File 1). After the initial tests, we changed from plastic recoverable to metal recoverable microdrives. In the first submission, we tested our full system (cap with metal, recoverable microdrive) in a mouse and a rat and we are continuously implanting animals using these newly developed methods. Since our submission, we have performed two chronic and two acute experiments in rats and two chronic experiments in mice.

Since our original submission, our lab has performed the following experiments.

**Table resptable1:** 

R_08	Long Evans	330	R	Plastic recov	Neuropixels from R07	Hippocampus	Chronic	Ongoing	NA	Homecage
R_09	Long Evans	345	R	Metal recov	Neuropixels	Hippocampus	Chronic	Ongoing	NA	Homecage linear maze
R_01A	Long Evans	430	NA	Metal recov	128-5 (DB) from M05	Hippocampus	Acute	NA	NA	NA
R_02A	Long Evans	430	NA	Metal recov	128-5 (DB) from R_01A	Hippocampus	Acute	NA	2 shank broke	NA
M_06	C57BL/6	31	M	Plastic recov	Micro-LED	Hippocampus	Chronic	Ongoing	NA	Homecage linear maze
M_07	C57BL/6	27	M	Plastic recov	Micro-LED	Hippocampus	Chronic	Ongoing	NA	Homecage linear maze
M_08	DBA/2J	34	M	Metal recov	128-5 (DB) from R02_A	Hippocampus	Chronic	Ongoing	NA	Homecage
M_09	C57BL/6	28	B	Metal recov	Assy-E1 (CN)	Hippocampus, entorhinal cortex	Chronic	Ongoing	NA	Homecage radial arm maze

2. Page 4, line 100 and table 1: What travel distance this refers to? The effective travel distance should be better explained. Once the probe is mounted in the shuttle, the travel distance is limited by the probe length and tip location from the microdrive base. Unexperienced users may misunderstood this important issue.

The travel distance refers to the overall distance that the arm can move from the top to the bottom of the drive body. The reviewer is right that travel distance depends on many factors, including length of shank(s), initial distance between arm and bottom of the drive body. We added a sentence to the revised manuscript for clarification.

3. Table 1: What are inclusion criteria? Are these microdrives designed for tetrodes or for silicon probes; mice only? Korshunov, 2006 is for single wires in a wide range of species and I fail to see how it could be repurposed for silicon probes. If the goal of the table is to summarize existing solutions, then it may fall short for including other microdrive designs. I would recommend focusing. Is Voroslakos et al., referring to this study ? Please, clarify.

Our inclusion criteria was the overall weight of the microdrive system, regardless of whether it uses silicon probes or wires. In the revised manuscript, we included microdrive systems that used silicon probes in mice. We also added an extra line in the legend explaining our inclusion criteria.

4. Figure 1D: the stereotaxic microdrive holder is very similar to that in Chung et al., (Figure 4). Is that the case? What are the slots in the attachment for?

We have been working with Sebastian Royer’s and Kamran Diba’s groups since 2017. They shared their design files which served as a basis of our microdrive and head cap developments. Our design is fundamentally the same as in Chung et al., and we make this clear in the text and describe the steps of improvements below.

Chung et al., 2016 design was designed for mice only. Hiroyuki Miyawaki has adapted Chung’s design for rats. We adapted Miyawaki’s design for mice (shorter travel distance) and made the following improvements over the years:

1. v01

a. Added nut to arm (reduced metal-to-plastic erosion).

2. v14

a. Updated shell component

i. Improved stability.

ii. Reduced footprint from 5.2x7.5 mm to 3.2x7.5mm. This enabled bilateral silicon probe implantation in the CA1 region of the hippocampus of rats (Rogers et al., 2021).

b. Reduced weight allowed us to use the same drive in mice (Valero et al., 2021) and rats. (We do not use the shorter mouse microdrive anymore).

3. v19

a. Added second male header pin to drive body (further improved stability).

4. v20

b. Increased thickness of the bottom part of the drive. Drive body became sturdier, less prone to bending.

5. v21

a. Improved arm design to overcome limitations of resin 3D printing. Most of the arms were not at 90-degree angles.

6. Metal prototype printed in plastic

a. Enables double silicon probe implantation in the mouse.

b. Nut is moved from the bottom of the drive to the top. This creates a flat surface at the bottom of the drive body making the shell-drive connection more stable.

7. 3D printed, metal recoverable microdrive.

5. Figure 2F-H and Figure 3D: In general, data is very descriptive and poorly relate with the potential added value of the system. I would consider providing some quantitative assessment to prove stability, or reusability, or consistency of some LFP feature between animals. What metrics is shown in Figure 2H and why is this useful to illustrate the added value of the microdrive? Same for Figure 2F.

We agree that Figure 2E and F did not illustrate the quality of the single cells well. It merely served to illustrate recordings of the single cells through the raw traces, approximated position and z-scored waveforms. Figure 2G and 2H show the quantitative measures of the single unit data. 2H illustrates the distribution of the refractory period violations in the recorded population (a common quality measure) and their spike amplitude. In the revised manuscript, we dedicate a full new figure (Figure 5) to the quantification of single cell features. We want to emphasize though that our head gear development is not primarily about long-term stability but for reusing silicon probes and head gear components with ease, decreasing the effective cost, experimental effort and complexity. The metal drive has the potential to allow for better long-term recordings, but we do not have firm quantitative data to support this, since this was not the goal of our experiments. For long-term recordings of multiple single units different technologies are needed (flexible probe recordings from the Frank group (Chung et al., 2019); Lieber group’s injectable electrode (Schoonover et al., 2020; Zhou et al., 2017)).

6. Introduction, line 49: authors make a good case for the emergence of novel integrated solutions such as uLED and microfluidic probes which benefit from the use of versatile headgear designs. However, both Wu et al. and Kim et al. seem to report on uLEDs only. Consider Altuna et al., Lab on Chip 2013 https://doi.org/10.1039/C3LC41364K of any other appropriate reference for integrated fluidic multi-site probes, or just avoid the mention to microfluidity.

In response to the Reviewer’s comment, we have removed reference to microfluidity from the revised manuscript because microfluidic probes require different solutions.

7. Introduction, lines 52-54: flexDrive, shuttledrive, etc. all of them are mostly dedicated to carry tetrodes. There are fewer designs for silicon probes, which may support the need for this paper, but authors avoid discussing the potential overlapping and added value between existing solutions and their own design. I feel the introduction will benefit from addressing this more sharply.

In our revised introduction, we address the different needs of tetrode and silicon drives. A disadvantage of tetrode recordings is that each tetrode needs to be moved separately.

8. Videos: Only two videos are provided to illustrate the head-cap systems of rats and mice. While documentation provided is useful, many relevant parts of the paper will strongly benefit from providing video support (e.g. microdrive assembly, implantation, recover).

In response to the Reviewer’s comment, we have created the following additional videos:

– Figure 1-video 1, showing how to assemble the recoverable metal microdrive.

– Figure 1-video 2, showing how to attach a Neuropixels probe to a recoverable, metal microdrive.

– Figure 2-video 1, showing how to assemble the mouse cap.

– Figure 3-video 1, showing how to assemble the rat cap.

We have also added the following two videos.

– Figure 4-video 1, showing how to recover a silicon probe from a mouse cap.

– Figure 4-video 2, showing how to recover a silicon probe from a rat cap.

9. Methods, line 352: stl files are provided via a Github. It would be useful to upload all of them together as supplementary material of the paper itself. Also, please, consider adding specific links to the different files repository in the Methods section.

Pending on acceptance of our revised manuscript, we will work with the editor to add our stl files as Supplementary material. We added the appropriate links in the Methods section.

10. Methods, line 369: authors recommend measuring probe impedance. This is not particularly easy with silicon probes. Please, add equipment information.

To collect electrophysiology data, we recommend using the RHD USB interface board from Intan (Intan Technologies LLC, CA, USA). This system enables users to measure the impedance of the attached device. For more details see:

https://intantech.com/files/Intan_RHD2000_eval_system.pdf page 14 Electrode Impedance Measurement section. This measurement provides a quick and rough estimate about the quality of the recording sites, whether they are below or above 2 MOhm (we can collect good wide-band signal if the impedance of the recording sites is below 2 MOhm). If higher precision is required, users can perform impedance measurement with NanoZ device (nanoZ Impedance Tester, Plexon Inc, TX, USA).

We added this information to the revised manuscript.

Reviewer #2 (Recommendations for the authors):This manuscript provides an updated guide on the procedures for performing chronic recordings with silicon probes in mice and rats in the lab of the senior author, who is one of the leaders in the use of this experimental method. The new set of procedures relies on metal and plastic 3D printed parts, and represents a major improvement over the older methodology (i.e. Vandecasteele et al. 2012).The manuscript is clearly written and the technical instructions (in the Methods section) seem rather detailed. The main concerns I had are as follows.

We thank the reviewer for carefully reading our manuscript and providing useful and constructive comments.

1. The present design is an improvement over Chung et al. (the most similar previously published explantable microdrive design, as far as I am aware) in terms of the footprint and travel distance. However, a main disadvantage of the system in its present form is that (apparently) it does not support Neuropixels probes. While such probes might not be suitable for some uses (e.g. to record from large populations in dorsal hippocampus), Neuropixels probes are of considerable interest to many labs.

Our microdrive and head cap system can also support Neuropixels probes. Since our initial submission, we have implanted a Neuropixels probe in the intermediate hippocampus of a rat using our recoverable, plastic microdrive. At the end of the experiment, the Neuropixels probe was successfully recovered, cleaned, and implanted again in a new rat. In addition, we designed a new arm for our metal microdrive which can support Neuropixels probes (Author response image 1) and implanted another rat (Author response image 2 and Figure 3—figure supplement 2).

We have also created a video showing how to attach Neuropixels probe to a metal microdrive (Figure 1-video 2).

**Author response image 1. sa2fig1:** Metal microdrive adapter for Neuropixels probe. A. Arm design for 64-channel silicon probes. 45^o^, front, side and top views are shown (from left to right). All dimensions are in mm. B. Changing the overall length (from 7.35 mm to 10 mm) and width (from 4 mm to 5.4 mm) of the 64-channel arm makes our metal microdrive compatible with Neuropixels probe. Note, that only three dimensions of the 64-channel arm were modified (red numbers). 45-degree, front, side and top views are shown (from left to right). All dimensions are in mm. C. Photograph of the different arm designs of the metal, recoverable microdrive (top shows an arm designed for a 64-ch silicon probe, bottom shows an arm designed for Neuropixels probe).

**Author response image 2. sa2fig2:** Recording of unit firing with Neuropixels probe attached to a metal microdrive in freely moving rat. A. Metal microdrive for Neuropixels probe (a – stereotax attachment, b – drive holder, c – metal microdrive, d – Neuropixels probe and e – Neuropixels headstage). B. Photo of Neuropixels probe attached to a metal microdrive (a-e same as in A). C. Location of probe implantation (Bregma – 4.8 mm, mediolateral + 4.6 mm, 11-degree angle). D. High pass filtered traces (1s) from a freely moving rat implanted with Neuropixels probe. Note the single unit activity in the cellular layer of cortex (top) and hippocampus (bottom).

2. The total weight of the mouse implant seems quite high (together with the headstage, I estimate it is >= 4gr). Could the authors provide the exact value, and describe whether this has any impact on the way the animal moves? Also, the authors should describe how the animals are housed (e.g. do they carry the headstage even when not being recorded). The authors say that a mouse can be implanted with more than one microdrive. The authors should clarify whether they actually have an experience with such implants, or is this just a suggestion based on their educated estimate?

The total weight of the metal microdrive, including the base, body and arm is 0.87 gram. Additional weight is the metabond and dental acrylic cement. The amount of cement that is used during surgery can vary between researchers and the type of surgery. The overall weight of the assembly also depends on the silicon probe with Omnetics connector(s) that is used for the surgery, e.g.: 32-channel micro-LED probe is 1.11g (NeuroLight Technologies LTD.), 64-channel 4-shank probe is 0.96g (ASSY E-1, Cambridge NeuroTech), 64-channel 5-shank probe is 1.05g (A5x12-16-Buz-Lin-5mm, NeuroNexus Ltd.) and a 128-channel 4-shank probe with integrated Intan chips is 0.94g (P128-5, Diagnostic Biochips). In addition, the overall weight of the entire assembly can change if optic fibers are used in optogenetic studies or if any custom connectors are implanted (e.g., connector and wires for brain stimulation). That is the reason why we reported the overall weight of each system (metal microdrive, mouse cap and rat cap) individually.

The implanted mice are single housed, and they do not carry the headstage while in the vivarium. During recordings, the headstage is attached and a counterbalanced pulley system ensures that the animal is not carrying the extra weight of the headstage. We have quantitatively compared running speed with traditional and the new head caps in both rats and mice (Figure 6).

The small footprint of the metal microdrive enables researchers to perform more than one silicon probe implantation in freely moving mice. For this purpose, larger mice (>35 g) are selected (Author response image 3).

**Author response image 3. sa2fig3:** Metal microdrive enables double silicon probe recordings in freely moving mice. A. Intraoperative photograph of double silicon probe implantation. Note that the metal microdrive on the left had been secured to the skull and the second drive is being implanted using the stereotaxic attachment and drive holder. The probe PCBs are placed on the copper mesh. B. Photograph focused on the metal microdrives.

3. There is no information in the Results section on the number of implants performed, the duration the animals were implanted, the quality of the recordings obtained, number of successes or failures. The figures merely provide examples of one successful recording in a mouse and in a rat. All these details should be provided, along with details of how many probes were reused and how many times (a brief mention of one case, lines 252-253 and 359-360, is not sufficient).

We have added Supplementary File 1 explaining all the details of our implants. We would like to refer the Reviewer to response #1 to Reviewer 1.

Adapting new technology is challenging. To date, we have extensive experience with the rat cap system only (n=3 users in the lab, n = 25 rats implanted). Two lab members have started to adapt our mouse cap and implanted 3 mice since our submission. We included their maze running behavioral data for comparison between the copper mesh and cap system.

Prior to the development of the metal microdrive, we have conducted an internal lab survey comparing the hand-made microdrive (Vandecasteele et al., 2012) and our recoverable, plastic microdrive. Six lab members who had extensive experience with both types participated (Figure 1—figure supplement 2). Our questions were:

1. On a scale 1-10, how would you compare the plastic, recoverable drive to the Vandecasteele et al., 2012 one in terms of: (a) ease of building a drive, (b) size and (c) ease of recovery.

Overall, the success rate of recovery is much higher using a recoverable microdrive system, but the size of the plastic, recoverable microdrive is limits certain experiments. This was one of the main motivations to develop the metal, recoverable microdrive.

4. In Figure 2, spike waveforms are classified as pyramidal, wide or narrow interneurons. I did not find any description of how this classification was performed.

We have removed the single cell putative cell types from the manuscript as this issue is not relevant to the current manuscript. Figure 2 has been simplified and a new figure 5 is dedicated to the single cell quantification.

5. Also in Figure 2, refractory period violations are reported in percent (permille in fact). First, it is not clear how refractory period was defined. Second, such quantification is incorrect in principle: we use refractory period violations to infer the rate of false positives. Yet the relationship between fraction of ISI violations and false positive rate depends on the firing rate of the neuron. For example, 0.1% of ISI violations is quite good for a unit spiking at 10 spikes/s, is so so for a unit spiking at 1 spike/s, and is very bad if the firing rate is 0.1 spike/s (see Hill et al. JNeurosci. 2011 for derivation). Alternatively,the authors can follow an approach described in an old paper by the same lab (Harris et al., JNeuropsysiol. 2000), quantifying the violations in spike autocorrelogram relative to its asymptotic height.

We have removed this panel from Figure 2 and dedicated a new figure (Figure 5) to the single cell quantification. Refractory violations can be used as an alarm for poor cluster quality. Absence of refractory violations alone does not guarantee good separation for the reasons the Reviewer mentioned.

6. Line 477: the authors write that the probes were mounted on a plastic microdrive. This seems to contradict the key claim of the manuscript (namely that the microdrives were from stainless steel).

We apologize if this description was not clear in the original manuscript. In the revised version, we have added a table (Supplementary File 1) explaining all details of each animal subject (species, strain, weight, cap type), type of silicon probe and microdrive used. As we explained in Response 3, our main goal was to test each system individually and once all components have been verified, we combined everything into one surgery.

The plastic and metal microdrives are based on the same principles. The implantation/recovery tools are also identical in design concepts. Based on our own experience, users dol not recognize any changes in terms of ease of use, ease of implantation and ease of recovery when changing from plastic recoverable microdrives to metal ones. The advantage of metal drives is size reduction, their multiple reusability and stability.

7. I believe that the work of Luo and Bondy et al., (eLife 2020) and should be references and compared to.

We reference Luo et al., (2020) in our revised manuscript. One of the main advantages of using a microdrive system is the ability to move the recording probe inside the brain tissue and sample new sets of neurons. This is not the case in Luo and Bondy et al., (*eLife* 2020).

Reviewer #3 (Recommendations for the authors):First, I'd like to applaud the authors for the development of a very clever device and for their clear description of how to use it. With the inclusion of more details around the compatibility of this device with specific probes, more support for specific claims in the manuscript, and more transparency about the success of probe recovery, this manuscript will inevitably serve as an important resource for many researchers.I'll leave one thought for consideration here before diving into specifics: the organization of this manuscript was a bit unclear to me. I think it makes sense to have a "Results section" which gives a high level description of the procedure and your design choices, as well as a "methods section" which outlines the protocol, but this needs to be clear. This could be solved by a line at the end of the introduction that says something like "Here we'll describe the results we obtained and a high level summary …. Readers can find detailed instructions in the methods as well as in the attached files…". I would also defer to the eLife editors for how they would like to handle this organization.The introduction of the paper is very well written, but towards the end there are several unsubstantiated claims. Specifically, the idea that this design reduces surgery time substantially. Can you be more specific, or back this up with timelines from other designs? On line 203 there is a reference to another paper after this claim, but this could be made more clear in the introduction. Which aspect of this procedure is quicker than other procedures? As I'll come to later in this review, there is also a claim about the recovery being "reliable" – knowing exactly how reliable, given your experience, would be extremely useful.

We thank the reviewer for carefully reading our manuscript and providing useful and constructive comments.

Overall, I have two suggestions that would greatly improve this manuscript.First, more neural data should be shown. In Figure 2 E-H and Figure 3, some neural data recorded using this device is shown, but this is not nearly enough for users to assess the usability of these probes. The mouse data is particularly sparse, and it is very difficult to make much out of Figure 3D without seeing isolated units. If 7 rats and 5 mice were recorded, more of this data should be shown. Specifically, users may be interested in seeing the stability of units over time, as well as the SNR levels on various days of recording. The methods there was daily recording – showing some of this data would be useful. How long has one of these devices been successfully used to record activity? These types of details are essential to ensuring that your device enables quality data collection.

We thank the Reviewer for these comments, which prompted us to add more quantifications and details. We added 3 new figures, 6 new videos and a Supplementary Table to address the Reviewer’s comment. Figure 5 is dedicated to the quantification of unit parameters.

Secondly, there should be a table detailing each procedure done with these devices, including the type of animal (mouse/rat), age (if available), sex, success of recording (and for how many days/weeks), and success of the probe recovery (beyond saying it is "highly successful", line 257). This comprehensive overview of exactly how reliable your device is will be very useful to readers.

As suggested, we have added Supplementary File 1, which list all critical technical details of all experiments included in the manuscript.

Several aspects of the manuscript could be clarified.1. The "footprint" of the device is given multiple times, but a height and width would also be useful.

The outer dimensions of the microdrive are specified in figure 1B. All dimensions of the drive are available in Table 1. We also added the width, height, and length to the main text.

2. There is a clear trade-off between the minimal wear and increased reusability of metal drives with the weight of these drives, and that should be acknowledged. Would it be possible to create such a drive with a sturdy but lighter plastic, and if not, why?

This is an important point. Stainless steel is about 100 times less flexible than plastic (Young's modulus of stainless steel: ~180 GPa vs plastic: ~2 GPa), yet the density is only 8 times higher (steel: ~8 kg/m3, plastic ~1 kg/m3). This is why the device can be made sturdier at a similar size. Manufacturing a plastic drive with the same strength would make the plastic drive substantially larger. While plastic drives are cheaper, their reusability is more limited. Both materials have their own advantages and weaknesses. However, reusability is an important factor since preparing a new head gear requires many hours of dedicated time.

3. Line 105 says the microdrive weighs 0.87 g – is this without any materials to attach it to the skull? Similarly, the overall weight of the entire assembly should be given.

The Reviewer is correct that the total weight of the metal microdrive, including the base, body and arm is 0.87 gram. Additional weight is the metabond and dental acrylic cement. The amount of cement that is used during surgery can vary between researchers and the type of surgery. The overall weight of the assembly also depends on the silicon probe with Omnetics connector(s) that is used for the surgery, e.g.: 32-channel micro-LED probe is 1.11g (NeuroLight Technologies LTD.), 64-channel 4-shank probe is 0.96g (ASSY E-1, Cambridge NeuroTech), 64-channel 5-shank probe is 1.05g (A5x12-16-Buz-Lin-5mm, NeuroNexus Ltd.) and a 128-channel 4-shank probe with integrated Intan chips is 0.94g (P128-5, Diagnostic Biochips). In addition, the overall weight of the entire assembly can change if optic fibers are used in optogenetic studies or if any custom connectors are implanted (e.g., connector and wires for brain stimulation). That is the reason why we reported the overall weight of each system (metal microdrive, mouse cap and rat cap) individually.

4. It is unclear what material the stereotaxic attachment is made of.

The stereotaxic attachment is made of plastic (clear v4 resin from FormLabs). We have added this information in the revised manuscript.

5. There doesn't seem to be any mention of how to actually attach the probe to the arm/shuttle. Is it glued? Which probes were used should be clear in the Results section (I see it is eventually mentioned in the methods). Relatedly, it should be clear which types of probes were tested with this microdrive, and which probes you would recommend using with it. Specifically, will this microdrive and assembly work with Neuropixels 1.0 probes? Being clear about which probe was used is especially important for the multi-probe implantation – presumably this will not work with probes with large PCB boards and/or headstages. Also, are multi-probe implantations possible in mice using your microdrives and assembly?

We attach the backend of the silicon probes to the arm using cyanoacrylate glue (Loctite, #45208). We have included this step in our Methods section, and we have also added this information to the figure legend of Figure 1 in the revised manuscript.

We added a new table summarizing which probes were used with our microdrive system. We have also performed the necessary modification of the arm design to support Neuropixels 1.0 probes (Figure 2) and performed a chronic surgery in a rat. Our microdrive can support any type of silicon probes provided that the arm is customized for the right size. We do not fully understand the Reviewer comment about “large PCB boards and/or headstages”. If the PCB boards and/or headstages are small/light enough for a rodent to carry them, our system will work with that device. Using our system with regular silicon probes, the headstages are attached to the animals during the recording session only. A counterbalanced pulley system makes sure that the animal is not carrying the extra weight of the headstage during recordings. In the revised manuscript, we provide quantitative data that compares the running speed of animals with various drives.

Dual probe implantations are possible with our system, as illustrated in Figure 5.

6. It is unclear how the headstage is affixed in either the rat or mouse assembly.

The headstage is attached to the Omnetics connector sitting on the PCB of the probe. A male header pin is attached to the probe PCB using cyanoacrylate glue and dental cement. During surgery, this male header pin is soldered to the cap system. There is no further need to support the headstage.

7. For probe recovery, it's important to note that distilled water will not be recommended for all probes. For example, neuropixels have very clear restrictions on what you should use with them. I would advise the reader accordingly. Tergazyme may be useful here as well.

The Reviewer is right that some probes might not be suitable for our cleaning procedure. In the revised manuscript, we clearly specified which probe types can be cleaned in the described way. We have also added a separate recommendation for cleaning Neuropixels 1.0 probes based on Luo et al. (2020) work.

8. How is grounding handled in these devices? There are multiple mentions of a skull screw used to affix the protective assembly, but in most designs, a skull screw is there to serve as a reference and/or grounding. (Sidenote: It is not clear to me why a ground screw is so bad for the animal, as is emphasized multiple times in the manuscript. We are also putting a large open whole in the skull…) Is there a ground wire in these devices? Similarly, is the copper mesh inside electrically connected to the probe, or is it kept isolated?

We either implant a 100-µm stainless-steel wire or a skull screw with an attached insulated wire (California Fine Wire, CA, USA) above the cerebellum to serve as ground for our recordings. After the silicon probe is implanted, this ground wire is soldered to the copper mesh or copper tape (mouse and rat cap, respectively). The probe’s ground and reference wires are also soldered to the copper mesh or copper tape. For the mesh to function as a Faraday cage, it must be grounded. Traditionally, the reference and ground electrodes were isolated from each other, but since we have been using Intan-based recording devices, we have not observed any difference in the quality of our recordings after shorting the reference and ground wires. The Intan headstages are shipped with shorted reference and ground (there is a zero-ohm jumper at R0 location on the printed circuit board of all headstages).

In the manuscript we refer to support skull screws as additional screws and not to the ground or reference screws. It has been common practice to use multiple support screws in rats and mice to attach the headgear to the skull. For instance, Vandecasteele et al., (2012) recommends using at least 4 support screws in rats. Recently, we eliminated this step and replaced the support skull screws with a surface bond material (Metabond). In our current protocol, we apply one or two layers of Metabond instead of multiple skull support screws, making the surgical procedure less invasive. We have also found that the animals recover faster. The bond is stable across the animal’s lifetime, and we have not observed any headgear loss with properly implanted animals.

9. Line 476 in methods is very unclear and mentions a plastic microdrive: "The probes ere mounted on a plastic recoverable microdrive to allow precise vertical movement after implantation (github.com/YoonGroupUmich/Microdrive) and implanted by attaching the base of the micro-drives to the skull with dental cement." Is this the same microdrive mentioned in the main manuscript? In general, it is unclear how the "Additional implantation information" relates to the main methods and seems that this information should be integrated into the other methods sections.

Please see our detailed answer to Comment 4 from Reviewer #1. Throughout the development of the metal microdrive and cap systems, we have gone through several iterations of the 3D printed plastic microdrive. This section in the manuscript was revised to better explain this history.

10. There is no mention of where to download the design files.

We have added links in the method section to our GitHub repository containing instructions and stl files: https://github.com/buzsakilab/3d_print_designs

We will create a separate zip file with all design files with the publication.

11. Data is provided for only 2 animals (one mouse, one rat) and 2 sessions – could more data be made available?

All our data will be made available after our manuscript is accepted for publication.

Pending on acceptance, we will work with the editors of *ELife* to link our data to this manuscript as well. Our lab has a long-lasting history of sharing our neuroscience data immediately after acceptance.

12. Code is provided for sorting (Kilosort Wrapper and phy plugin).

We used KiloSort together with KilosortWrapper:

https://github.com/petersenpeter/KilosortWrapper

Phy1 and phy2 plugins: https://github.com/petersenpeter/phy1-plugins

https://github.com/petersenpeter/phy2-plugins

[Editors' note: further revisions were suggested prior to acceptance, as described below.]

Reviewer #2 (Recommendations for the authors):I would like to encourage the authors to incorporate into the final version of the paper all the relevant technical details from the rebuttal. For example, in their response the authors mention using a pulley to counterbalance the headstage (yet this seems not to be mentioned in the manuscript), and similarly with preferentially using large (>35gr) mice.

We have added this information to the manuscript.

Reviewer #3 (Recommendations for the authors):There are several points that should be addressed within the revised manuscript:The new Figure 5 illustrates the use of two different recording devices in mice. Figure 5A demonstrates the units recorded over time on each shank as the probe is lowered each day. However, it is unclear to me how Figure 5B relates to 5A – in 5B it looks as if all four shanks are at the same depth, whereas in 5A they are at different depths. Ultimately, it seems like some sort of integration representation of these two panels where viewers can appreciate the location of single units on each shank over days would be the most useful. On this same figure, the axis labels for 5F and 5I are a bit misleading, because this is not about the noise level of the recording, but about the waveform. I'd suggest changing the figure and the wording in the text to "Waveform relative noise level" so that readers do not confuse this with overall signal to noise in the recording. The axis on 5G could be readjusted so that readers can appreciate the data. In the figure caption, I'd suggest spelling out "ACGs" for readers unaccustomed to that shorthand. Presumably these ACGs are from the unit in the box? If so, I'd clarify in the caption.

We apologize for this confusion. Indeed, Figure 5A captures the single unit stability visually with individual shanks plotted above each other and panel B shows the correct horizontal layout and captures the distribution across days for units across all shanks. We have altered the figure legend to better describe this and switched the subpanels in B. We have added “Waveform relative noise level” to the appropriate labels. We also updated the figure caption for J-L. It reads now: “J-L Neuropixels probe recording, where the same putative interneuron was tracked across four days. J Average waveforms (bandpass filtered 300-10000 Hz) of a putative interneuron recorded on 16 channels across 4 days (left). The average waveforms recorded at the site with the largest amplitude waveform is highlighted on the right (waveforms are color-coded by recording day). Autocorrelation histograms (K) and spike amplitudes (L; from Kilosort) for the same single unit, color-coded by recording day.”

Related to Figure 5, the corresponding text says, "suggesting either that the distance between the electrode sites and neuron bodies decreased or that large size neurons were recorded," but it is important to note that the probe was being moved into different brain areas over these days of recording. The text should be modified to clarify this – the clear difference from day 5 to the other days is almost definitely explained by the movement to deeper brain structures. This paragraph should also note the clear decrease in the # of units with the reimplanted probe, as well as the clear increase in the noise level in the reimplanted probe.

In response to the Reviewer’s suggestion, we added this information to the revised manuscript. We would also like to point out that despite a successful probe recovery, the recording quality can deteriorate over time reducing the number of high-quality single unit clusters (e.g., increased impedance of recording sites and decreased signal-to-noise over time).

The authors have also added behavioral data, shedding light on the ability of animals to move with this headgear, however some clarity around these behavioral findings is needed. On line 219 it reads, "The 3D printed head cap system is comparable in weight to manually built headmounts" which is an abrupt and unclear transition to the paragraph about the impact of the headgear on behavior – reading between the lines, I think the authors are saying, "… however, we wanted to verify still that our headgear would not impede the animal's behavior." It is also unclear from this paragraph how this behavior was measured. The methods section describes mice and rats running on a track – were they headfixed? It is also unclear how the water reward is relevant here – did animals need to collect a water reward in order for the track to continue moving? Also, the mice with 3D printed headgear ran on a different track than the mice with manually printed headgear – some mention of this in the main text is warranted for transparency in interpreting the behavioral results shown. The wording in the methods is also unclear – is the track circular or a figure eight? Finally, contextualizing the results found here in terms of typical speeds on such a treadmill (e.g., in the discussion, see the point below) would be very useful to readers.

The Reviewer is correct that we wanted to test whether our new headgear system would interfere with the behavior of freely moving mice and rats. We added the suggested sentence to the revised manuscript. All animals were freely moving, not head fixed. Rats and mice were water deprived and had to collect water as reward. The linear mazes had ‘reward areas’ on each end where reward was delivered via an automatic infrared-beam triggered system. Animals only received water reward for trials in which they travelled from one reward site to the other. We clarified the maze design in the manuscript: ”120-cm diameter circular track with a diagonal path allowing the animals to run in a figure-eight pattern”, its similar to a classical figure-eight maze, yet return arms are altered to a continuous circle, so that mice make fewer sharp turns. The animals are water deprived and earn reward when performing the alternation task correctly. The water reward was given at the start of the central path. We added this information to the Methods section.

In this same paragraph, the authors state they found a "small significant difference," but this wording is misleading. A statistical test result is or is not significant, and the authors should remove the word small. Readers can determine whether or not the absolute value of the p-value is informative. There is a typo in this section also: "We also performed the same test on mice subjects and found a small significant difference between the median running speed of the two *rat* groups (KS-test; p = 0.045) but no significant different between the 95 percentile speeds (KS-test; p = 0.24)." I would also recommend that the authors write out the full name of the KS test, at least on the first mention.

We agree with the Reviewer that a statistical test result is either significant or not. We have removed “small” from the revised manuscript. We also spell out Kolmogorov-Smirnov test (KS-test) when it is first mentioned. The typo was also corrected.

Line 139 says, "High-quality electrophysiological signals can be collected from freely moving mice for weeks and months (Figure 2E and F)" however it is unclear from this figure whether this data was indeed recorded over weeks and months. If this is a claim the authors cannot substantiate with data, it should be removed. Figure 5 does indeed show "weeks" with the headgear (showing data out to day 16 for the 4-shank probe and 17 for neuropixels), so it seems like an applicable figure to reference here, without the mention of "months." Similar changes should be made for the reference to the rat recordings and Figure 3 – it's unclear based on the figure caption how long after implant these extracellular traces were obtained.

We have removed the “months” from the revised manuscript and added the requested information (post-op day 18) to the caption of Figure 3.

Finally, although these are significant additions to the main portion of the text, there is no mention of either the single-unit figure or the behavioral results in the discussion. A contextualization of these data in terms of the headgear's usability as well as a comparison of these findings to other recoverable drives would be useful and warrant discussion.

In response to the Reviewer’s comment, we included a new paragraph in the Discussion.